# Learning the intrinsic dynamics of spatio-temporal processes through Latent Dynamics Networks

Francesco Regazzoni [1] ✉, Stefano Pagani[1], Matteo Salvador[1,2], Luca Dede'[1] & Alfio Quarteroni[1,3]

Predicting the evolution of systems with spatio-temporal dynamics in response to external stimuli is essential for scientific progress. Traditional equations-based approaches leverage first principles through the numerical approximation of differential equations, thus demanding extensive computational resources. In contrast, data-driven approaches leverage deep learning algorithms to describe system evolution in low-dimensional spaces. We introduce an architecture, termed Latent Dynamics Network, capable of uncovering low-dimensional intrinsic dynamics in potentially non-Markovian systems. Latent Dynamics Networks automatically discover a low-dimensional manifold while learning the system dynamics, eliminating the need for training an auto-encoder and avoiding operations in the high-dimensional space. They predict the evolution, even in time-extrapolation scenarios, of space-dependent fields without relying on predetermined grids, thus enabling weight-sharing across query-points. Lightweight and easy-to-train, Latent Dynamics Networks demonstrate superior accuracy (normalized error 5 times smaller) in highly-nonlinear problems with significantly fewer trainable parameters (more than 10 times fewer) compared to state-of-the-art methods.

Mathematical models based on differential equations, such as Partial Differential Equations (PDEs) and Stochastic Differential Equations (SDEs), can yield quantitative predictions of the evolution of space-dependent quantities of interest in response to external stimuli. Pivotal examples are given by fluid dynamics and turbulence[1], wave propagation phenomena[2], the deformation of solid bodies and biological tissues[3], molecular dynamics[4], price evolution of financial assets[5], epidemiology[6]. However, the development of traditional modeling-and-simulation approaches carry several mathematical and computational challenges. Model development requires a deep understanding of the physical processes, the adoption of physics first principles or empirical rules, and their translation into mathematical equations. The values of parameters and of boundary and initial conditions required to close the model are often unknown, increasing the intrinsic dimensionality of the solution space. Finally, the computational cost

that accompanies the (possibly many-query) numerical approximation of such mathematical models may be prohibitive and hinder their use in relevant applications[7,8].

In recent years, we are witnessing the introduction of a new paradigm, namely data-driven modeling[9–15], as opposed to traditional physics-based modeling, enabled by recent advances in optimization, high-performance computing, GPU-based hardware, artificial neural networks (NNs) and Machine/Deep Learning in general. Data-driven modeling methods hold promise in overcoming the limitations of traditional physics-based models, either as a replacement for them or in synergy with them[16,17]. On the one hand, data-driven techniques are employed to learn a model directly from experimental data[9,10]. On the other hand, instead, they are used to build a surrogate for a high-fidelity model – the latter being typically based on the numerical approximation of systems of differential equations – from a dataset of

[1]MOX, Department of Mathematics, Politecnico di Milano, Milan, Italy. [2]Institute for Computational and Mathematical Engineering, Stanford University, Stanford, CA, USA. [3]École Polytechnique Fédérale de Lausanne, Lausanne, Switzerland. ✉e-mail: francesco.regazzoni@polimi.it

precomputed high-fidelity simulation snapshots[14,16]. This paradigm is successful in many-query contexts, that is when the computational resources spent in the offline phase (generation of the training data and construction of the data-driven surrogate model) are repaid by a large number of evaluations of the trained model (online phase), as is the case of sensitivity analysis, parameter estimation and uncertainty quantification. Another case of interest is when real-time responses are needed, like, e.g., in clinical scenarios[18].

Several methods have been recently proposed for automatically learning the dynamics of systems exhibiting spatio-temporal behavior[12,19–23]. Typically, these methods discretize the space-dependent output field into a high-dimensional vector (e.g., by point-wise evaluation on a grid, or by expansion with respect to a Finite Element basis or to a Fourier basis) and then compress it by means of dimensionality reduction techniques, based e.g. either on proper orthogonal decomposition (POD) of a set of snapshots[7,24–26], or on fully connected auto-encoders, or else on convolutional auto-encoders[19,20,27–30]. The underlying assumption is that the dynamics can be represented by a limited number of state variables, called latent variables, whose time evolution is learned either through NNs with recurrent structure (such as RNNs[31], LSTMs[20,29] or ODE-Nets[32]), dynamic mode decomposition[27], SINDy[12,33], fully-connected NNs (FCNNs)[30], or DeepONets[19].

When a high-fidelity model is available, there are also techniques for building reduced-order models by exploiting knowledge of the equations[34–41]. These latter methods are however intrusive, unlike the formers, which learn a model in a data-driven manner using only a dataset of input-output pairs. Intrusive techniques are typically based on projecting the high-fidelity model into a low-dimensional space, obtained by POD or by greedy algorithms. In the case of nonlinear models, however, such techniques require special arrangements, such as the (discrete) empirical interpolation method[42–44], but this entails a difficult trade-off between accuracy and computational cost. Furthermore, many problems feature a slow decay of the Kolmogorov $n$-width, an index of the amenability of the solution manifold to be approximated by an $n$-dimensional linear subspace[45,46]. In many cases of interest, such as advection-dominated problems or high Reynolds number flow equations, POD-based methods achieve reasonable accuracy only for high values of $n$[28]. This limits their use in practical applications.

In this paper, we introduce a family of NNs, called Latent Dynamics Networks (LDNets), that can effectively learn, in a data-driven manner, the temporal dynamics of space-dependent fields and predict their evolution for unseen time-dependent input signals and unseen scalar parameters. LDNets automatically discover a compact encoding of the system state in terms of (typically a few) latent scalar variables. Remarkably, the latent representation is learned without the need of using an auto-encoder to explicitly compress a high-dimensional discretization of the system state. Furthermore, LDNets are based on an intrinsically space-dependent reconstruction of the output fields. Indeed, instead of yielding a discrete representation of the fields (e.g. point values on a spatial mesh), LDNets are able to generate output fields defined at any point of space, in a meshless manner. As a consequence, the (typically high-dimensional) discrete representation of the output is never explicitly constructed. These features make the training of LDNets extremely lightweight, and boost their generalization ability even in the presence of few training samples and even in time-extrapolation regimes, that is for longer time horizons than those seen during training.

We denote by $\mathbf{y} : \Omega \times [0, T] \to \mathbb{R}^{d_y}$ an output field we aim to predict, where $\Omega \subset \mathbb{R}^d$ is the space domain and $T > 0$ is the final time. The evolution of $\mathbf{y}$ is driven by the input $\mathbf{u} : [0, T] \to \mathbb{R}^{d_u}$, that is, a set of time-dependent signals or, more simply, constant parameters. Our goal is to unveil, starting from data, the laws underlying the dependence of $\mathbf{y}$ on $\mathbf{u}$. We denote by $\mathcal{S}_{\text{train}}$ the set of training samples. For each $i \in \mathcal{S}_{\text{train}}$, we assume to have available some observations of $\mathbf{u}^i(\tau)$ and of $\mathbf{y}^i(\boldsymbol{\xi}, \tau)$, sampled at a finite set of points $\boldsymbol{\xi} \in \Omega$ and times $\tau \in [0, T]$, originating, for example, from a collection of sensors.

An important (albeit not exclusive) example is the case when the dynamics we aim to learn underlies a differential model in the form of

$$\begin{cases} \partial_t \mathbf{z}(\mathbf{x}, t) = \mathcal{F}(\mathbf{z}(\mathbf{x}, t), \mathbf{u}(t)) & \text{in } \Omega \times (0, T] \\ \mathbf{y}(\mathbf{x}, t) = \mathcal{G}(\mathbf{z}(\mathbf{x}, t), \mathbf{x}) & \text{in } \Omega \times (0, T] \\ \mathbf{z}(\mathbf{x}, 0) = \mathbf{z}_0(\mathbf{x}) & \text{in } \Omega \end{cases} \quad (1)$$

where $\mathbf{z}(\mathbf{x}, t)$ is the state variable, $\mathcal{F}$ is a differential operator and $\mathcal{G}$ is the observation operator. Meaningful examples are provided in the Results section. In particular, LDNets can be also used to generate a reduced-order model of (1), by passing through data generated via a numerical approximation of (1), e.g. by the Finite Element method, called full-order model (FOM).

An LDNet consists of two sub-networks, $\mathcal{NN}_{\text{dyn}}$ and $\mathcal{NN}_{\text{rec}}$, that is two FCNNs with trainable parameters $\mathbf{w}_{\text{dyn}}$ and $\mathbf{w}_{\text{rec}}$, respectively (see Fig. 1). The first NN, namely $\mathcal{NN}_{\text{dyn}}$, evolves the dynamics of the latent variables $\mathbf{s}(t) \in \mathbb{R}^{d_s}$ according to the differential equation

$$\dot{\mathbf{s}}(t) = \mathcal{NN}_{\text{dyn}}(\mathbf{s}(t), \mathbf{u}(t); \mathbf{w}_{\text{dyn}}) \qquad \text{in } (0, T]. \quad (2)$$

We remark that, thanks to the hidden nature of $\mathbf{s}(t)$, we can assume without loss of generality the initial condition $\mathbf{s}(0) = \mathbf{0}$ (see[47] for a discussion on this topic in a similar framework). The inputs of $\mathcal{NN}_{\text{dyn}}$ are the latent states $\mathbf{s}(t)$ and the input signal $\mathbf{u}(t)$ at the current time $t$. Instead, the second NN, $\mathcal{NN}_{\text{rec}}$, is used to reconstruct $\widetilde{\mathbf{y}}$, an approximation of the output field $\mathbf{y}$ at any time $t \in [0, T]$ and at any query point $\mathbf{x} \in \Omega$:

$$\widetilde{\mathbf{y}}(\mathbf{x}, t) = \mathcal{NN}_{\text{rec}}(\mathbf{s}(t), \mathbf{x}; \mathbf{w}_{\text{rec}}) \qquad \text{in } \Omega \times (0, T]. \quad (3)$$

We remark that the reconstruction network $\mathcal{NN}_{\text{rec}}$ is independently queried for every point $\mathbf{x} \in \Omega$ for which the solution is sought. The input signal $\mathbf{u}(t)$ can optionally be given as input to the reconstruction network $\mathcal{NN}_{\text{rec}}$ (an example is given in Test Case 2). Normalization layers are employed to facilitate training. They are defined to guarantee that each feature approximately spans the interval $[-1,1]$. We also normalize the time variable, by dividing the time steps by a characteristic time scale $\Delta t_{\text{ref}}$, considered as an hyperparameter of the model. In case an output feature has a long-tailed distribution, we supplement the normalization layer with a non-trainable nonlinear layer to compress the tails. See Methods for further details.

Optionally, the architecture of $\mathcal{NN}_{\text{dyn}}$ and $\mathcal{NN}_{\text{rec}}$ are adapted to enforce a-priori knowledge. On the one hand, in case data are collected starting from an equilibrium configuration associated with the input $\mathbf{u}_{\text{eq}}$, we define $\mathcal{NN}_{\text{dyn}}$ as

$$\mathcal{NN}_{\text{dyn}}(\mathbf{s}, \mathbf{u}; \mathbf{w}_{\text{dyn}}) = \widetilde{\mathcal{NN}}_{\text{dyn}}(\mathbf{s}, \mathbf{u}; \mathbf{w}_{\text{dyn}}) - \widetilde{\mathcal{NN}}_{\text{dyn}}(\mathbf{0}, \mathbf{u}_{\text{eq}}; \mathbf{w}_{\text{dyn}}),$$

where $\widetilde{\mathcal{NN}}_{\text{dyn}}$ is a trainable FCNN, thus enforcing by construction the equilibrium condition. On the other hand, if the value of the output fields is a priori known on a subset of the domain $\Omega$, we define $\mathcal{NN}_{\text{rec}}$ as

$$\mathcal{NN}_{\text{rec}}(\mathbf{s}, \mathbf{u}, \mathbf{x}; \mathbf{w}_{\text{rec}}) = \mathbf{y}_{\text{lift}}(\mathbf{x}) + \widetilde{\mathcal{NN}}_{\text{rec}}(\mathbf{s}, \mathbf{u}, \mathbf{x}; \mathbf{w}_{\text{rec}})\psi(\mathbf{x}),$$

where $\widetilde{\mathcal{NN}}_{\text{rec}}$ is a trainable FCNN, $\mathbf{y}_{\text{lift}}$ is the lifting of the value to be prescribed, that is an extension to the whole domain, and $\psi : \Omega \to \mathbb{R}$ is a mask, that is a smooth function vanishing on the region where the output is prescribed. See Methods for further details.

The two NNs, $\mathcal{NN}_{\text{dyn}}$ and $\mathcal{NN}_{\text{rec}}$, are simultaneously trained via empirical risk minimization, that is by minimizing the quadratic

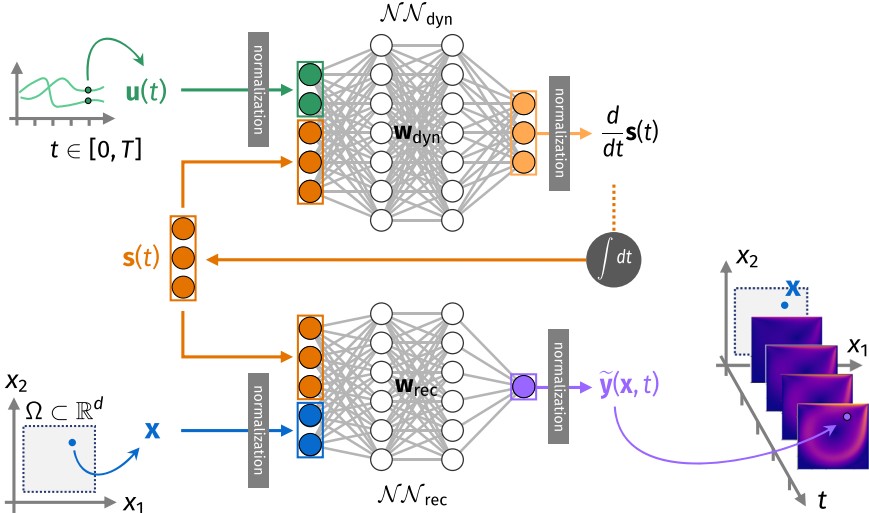

**Fig. 1 | LDNet architecture.** The network $\mathcal{NN}_{dyn}$ receives the input $\mathbf{u}(t)$ and the latent state $\mathbf{s}(t)$ and returns the time derivative of the latent state, thus defining its dynamics. The network $\mathcal{NN}_{rec}$, instead, is evaluated only when an estimate of the output field $\mathbf{y}$ is sought. More precisely, an approximation of $\mathbf{y}(\mathbf{x}, t)$ is recovered by giving as an input to $\mathcal{NN}_{rec}$ the latent state at time $t$ and the query space coordinate $\mathbf{x} \in \Omega$. In general, the reconstruction network $\mathcal{NN}_{rec}$ might take as an input $\mathbf{u}(t)$ as well (see e.g. Results, Test Case 2); for simplicity, in the figure we represent the special case when $\mathcal{NN}_{rec}$ does not depend on $\mathbf{u}(t)$.

difference between predictions and observations on the training dataset. Tikhonov regularization on the NNs' weights is employed to mitigate overfitting. Thanks to the simultaneous end-to-end training of the two NNs, the latent space is discovered at the same time as learning the dynamics of the system. This generalizes the approach presented in[47] for the case of time signals as outputs.

To train the parameters, we employ a two stage strategy, consisting of a few hundreds epochs of the Adam optimizer[48], followed by the BFGS algorithm[49]. BFGS is more accurate than Adam, but more prone to get stuck in local minima, which is why it is useful to precede it with some Adam iterations, which provide a good initial guess. To tune hyperparameters, we employ a Bayesian approach, namely the Tree-structured Parzen Estimator algorithm[50], combined with Asynchronous Successive Halving scheduler to early terminate bad hyperparameters configurations[51].

Further details about the time discretization of (2), features normalization, parameters training and hyperparameter tuning are provided in Methods.

## Results

We demonstrate the effectiveness of LDNets through several test cases. First, we consider a linear PDE model to analyze the ability of LDNets to extract a compact latent representation of models that are progressively less amenable to reduction. Then, we consider the time-dependent version of a benchmark problem in fluid dynamics. Finally, we compare LDNets with state-of-the-art methods in a challenging task, that is, learning the dynamics of the Monodomain equation coupled with the Aliev-Panfilov model[52], a highly non-linear excitation-propagation PDE model used in the field of cardiac electrophysiology modeling, of which we consider both a one-dimensional and a two-dimensional version. For more details on the test cases and on the results, we refer the interested reader to SI.

We focus on synthetically generated data obtained by numerical approximation of differential models, thus allowing us to test LDNet predictions against ground-truth results. We evaluate the prediction accuracy of the trained models using two metrics: the normalized root-mean-square error (NRMSE) and the Pearson dissimilarity, $1 - \rho$, where $\rho$ is the Pearson correlation coefficient.

### Test Case 1: advection-diffusion-reaction equation

We consider the linear advection-diffusion-reaction (ADR) equation on the interval $\Omega = (-1, 1)$:

$$\frac{\partial z(x, t)}{\partial t} - \mu_1 \frac{\partial^2 z(x, t)}{\partial x^2} - \mu_2 \frac{\partial z(x, t)}{\partial x} + \mu_3 z(x, t)$$
$$= f(x, t) \quad x \in (-1, 1), t \in (0, T]. \tag{4}$$

This PDE is widely used, e.g., to describe the concentration $z(x, t)$ of a substance dissolved in a channel[53]. The constant parameters $\mu_1, \mu_2$ and $\mu_3$ respectively represent diffusion, advection and reaction coefficients, while the forcing term $f(x, t)$ is a prescribed external source. We consider an initial condition $z(x, 0) = z_0(x)$ and periodic boundary conditions.

To generate the training dataset, we employ a high-fidelity FFT-based solver on 101 equally spaced grid points, combined with an adaptive-time integration scheme for stiff problems[54,55]. Then, we subsample the time domain in 100 equally distributed intervals. We challenge LDNets in predicting the space-time evolution of the target variable $\mathbf{y}(x, t) = z(x, t)$ by considering three cases of increasing complexity (Test Cases 1a, 1b, 1c), in which the input $\mathbf{u}$ is associated either with the parameters $\mu_1, \mu_2$ and $\mu_3$, or with the forcing term $f(x, t)$.

### Test Case 1a: finite latent dimension, constant parameters

First, we consider $z_0(x) = \cos(\pi x)$ and $f \equiv 0$. We aim at predicting the evolution of $z(x, t)$, depending on the constant parameters $\mathbf{u}(t) \equiv (\mu_1, \mu_2, \mu_3)$. Due to the linearity of the equation, the solution is, at any time $t$, a sine wave with period 2, and can be thus unambiguously identified by two scalars (namely, the wave amplitude and phase, or equivalently, the real and imaginary part of the Fourier transform at frequency 0.5). In other terms, the intrinsic dimension of the solution manifold is strictly equal to 2. This provides therefore an ideal testbed for the capability of LDNets to recognize and learn a low-dimensional encoding of the system state from data.

The LDNet, trained on 100 samples, achieves an excellent accuracy when tested on 500 unseen samples. Indeed, the NRMSE is $1.88 \cdot 10^{-5}$ on the test set, against a training NRMSE of $1.81 \cdot 10^{-5}$. Pearson dissimilarity is $3.30 \cdot 10^{-9}$ on the test set and $3.00 \cdot 10^{-9}$ on the training set. The very small differences in the accuracy metrics between training and test sets provide evidence that the

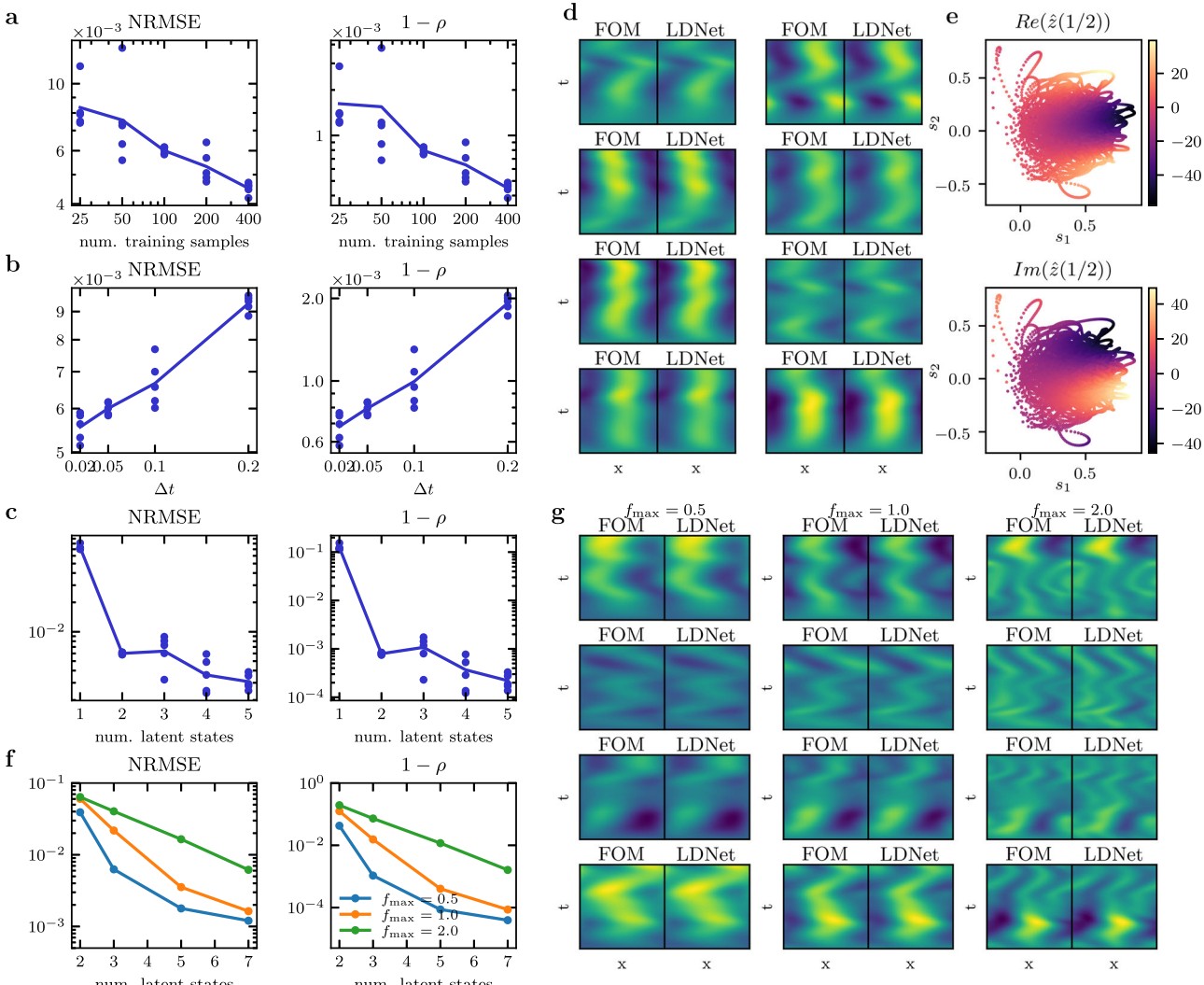

**Fig. 2 | Results of Test Case 1. a–c** Testing accuracy of Test Case 1b, as a function of the number of training samples (with $\Delta t = 0.05$ and 2 latent variables), of $\Delta t$ (with 100 training samples and 2 latent variables), and of the number of latent variables (with $\Delta t = 0.05$ and 100 training samples). For each setting we perform 5 training runs with random weights initialization. Each dot corresponds to a training run, while the solid line is the geometric mean. **d** FOM against LDNet predictions on 8 testing samples for Test Case 1b. The abscissa corresponds to space and the ordinate to time. **e** Mapping from the latent space trajectories and the Fourier space coefficients of the FOM solution for the testing samples of Test Case 1b. **f** Testing accuracy of Test Case 1c as a function of the number of latent states and of the maximum input frequency $f_{max}$. **g** FOM against LDNet predictions on 4 testing samples for Test Case 1c, obtained by employing 5 latent states, for different maximum input frequencies (reported above the figure).

trained LDNet reproduces the FOM dynamics with great fidelity and without overfitting, that is with remarkably good generalization capabilities.

**Test Case 1b: finite latent dimension, time-dependent inputs**
We now consider the case of time-dependent inputs, with a forcing term $f(x, t) = u_1(t) \cos(\pi x - u_2(t))$, where $\mathbf{u}(t) = (u_1(t), u_2(t))$ represents an input signal that can vary in time within a bounded set. Similarly to Test Case 1a, the solution manifold has dimension 2 (thanks to the equation being linear and to the forcing term having constant frequency), but learning the dynamics becomes more challenging due to the presence of time-dependent inputs.

We test the accuracy of LDNets for an increasing number of training samples, ranging from 25 to 400 (Fig. 2a). Remarkably, LDNets generalize well to unobserved samples even for a very small number of training samples, such as 25. As desirable, the accuracy of predictions improves as the time discretization step size is reduced and as the number of training samples increases (Fig. 2b). Indeed, because of the non-intrusive nature of LDNets,

their ability to discover system dynamics is limited by the information contained in the training set, as it is common in data-driven model reduction/discovery methods[22,23,47]. Still, as the input space is covered more densely, LDNets are able to leverage that information as attested by the significantly decreasing test error.

In this test case, despite the FOM state is discretized using 101 space points, the intrinsic dimension of the solution manifold is much lower, namely 2. In fact, a compact representation of the system state is obtained by means of the Fourier transform at frequency 0.5 (denoted by $\hat{z}(0.5)$) and consists of two scalars (i.e. $Re(\hat{z}(0.5))$ and $Im(\hat{z}(0.5))$). Therefore, we test whether LDNets can discover an equivalent encoding of the system state: in Fig. 2e, we plot $Re(\hat{z}(0.5))$ and $Im(\hat{z}(0.5))$ along the testing trajectories in the latent space $(s_1, s_2)$. In this figure a well-defined mapping emerges from the two latent states to the two Fourier coefficients: LDNets are capable of discovering a compact encoding based on an operator that is equivalent to the Fourier transform, without being explicitly instructed to do so, that is in a fully data-driven manner.

Finally, we train LDNets for increasing number of latent variables, ranging from 1 to 5 (Fig. 2c). As expected, the prediction accuracy significantly drops when going from 1 to 2 latent variables, that is when the intrinsic solution manifold dimension is reached. With more than 2 latent variables, it reaches a plateau, showing only a slight decrease due to increased model capacity, accompanied by a larger variance in the output.

## Test Case 1c: infinite latent dimension

Finally, we consider a forcing term $f(x, t) = u_1(t) \cos(2\pi u_3(t)x - u_2(t))$, where $\mathbf{u}(t) = (u_1(t), u_2(t), u_3(t))$ is the time-dependent input signal. The forcing frequency $u_3(t)$ varies within an interval $[0.25, f_{\max}]$. Hence, the solution manifold of (4) has a potentially infinite dimension, being $z$ the superimposition of a continuum of frequencies. Still, the results show that LDNets are able to discover effective low-dimensional encodings of the state.

First, we set $f_{\max} = 0.5$ and we train LDNets for increasing number of latent states, from 2 to 7. Remarkably, as the number of latent states increases, LDNets discover more effective encodings, that reflect in an increasing prediction accuracy (Fig. 2f, blue line). Unlike Test Case 1b, where the intrinsic size of the solution manifold is 2 and this leads to a stagnation of the error, here the error decreases significantly even for higher numbers of latent variables. Still, for higher numbers of latent variables, we have a slowdown in the decreasing trend of the error, due to two factors: on the one hand, the finite size (100 samples) of the training set (see in this regard Fig. 2a), on the other hand, the optimizer that may not find the global minimum of the loss function. By increasing $f_{\max}$ to 1 and 2, the FOM state gets less prone to be represented by a compact encoding, since the spectrum of the solution is wider. Prediction accuracy is indeed lower than in the case $f_{\max} = 0.5$, but it improves greatly by increasing the number of latent variables.

To further assess the crucial role of including latent states in the model, we analyze the results obtained by removing the latent variables, namely by considering an ODE-Net fed by the input signal and the query point, and tracking the evolution of the output at the considered point (see SI for more details). We train this architecture by considering Test Case 1c, with $f_{\max} = 0.5$. To ensure a fair comparison, we employ the same dataset and the same hyperparameter tuning algorithm used for LDNets. The results (see Fig. 3) reveal that without latent states the prediction accuracy of the model is significantly reduced. Moreover, the greater the number of latent states, the greater the ability of the model to capture finer and finer features of the dynamics. We conclude that the presence of a latent state is a crucial architectural choice for LDNets. The latent variables allow nonlocal information to propagate across the computational domain $\Omega$. With the architecture considered in this comparison, instead, the solution evolves in each point unaware of the state of surrounding points, despite the point coordinate is provided to the ODE-Net. Conversely, LDNets are able to learn systems whose dynamics is determined by spatial correlations. Notable examples are provided in the next sections.

## Test Case 2: unsteady Navier–Stokes

The 2D lid-driven cavity is a well-known benchmark problem in fluid dynamics[56], which may exhibit a wide range of flow patterns and vortex structures when increasing the Reynolds number. We challenge LDNets in learning an unsteady version of the lid-driven cavity problem, where the velocity prescribed on the lid $\Gamma_{\text{top}}$ (the top portion of the boundary) is a time-dependent input $u(t)$ (see Fig. 4a). During the simulations, the Reynolds number varies over time by reaching peaks of nearly 1500. This problem is challenging also because of discontinuities in the velocity field at the two top corners. The goal here is to predict the velocity field (that is, we set $\mathbf{y}(\mathbf{x}, t) = \mathbf{v}(\mathbf{x}, t)$) for each

prescribed $u(t)$:

$$
\begin{aligned}
\rho \frac{\partial \mathbf{v}}{\partial t} + \rho(\mathbf{v} \cdot \nabla)\mathbf{v} - \mu \Delta \mathbf{v} + \nabla p = \mathbf{0} & \quad \mathbf{x} \in \Omega, t \in (0, T], \\
\nabla \cdot \mathbf{v} = 0 & \quad \mathbf{x} \in \Omega, t \in (0, T], \\
\mathbf{v} = u(t)\mathbf{e}_x & \quad \mathbf{x} \in \Gamma_{\text{top}}, t \in (0, T], \\
\mathbf{v} = \mathbf{0} & \quad \mathbf{x} \in \partial\Omega \setminus \Gamma_{\text{top}}, t \in (0, T], \\
\mathbf{v} = \mathbf{0} & \quad \mathbf{x} \in \Omega, t = 0,
\end{aligned} \tag{5}
$$

where the dependence of the velocity $\mathbf{v}$ and pressure $p$ on space and time is understood. As shown in[57], a simple quadratic loss function is not adequate for capturing small vortex structures, because of their small impact, compared to medium- and large-scale structures, on the loss function. Therefore, we use the following goal-oriented metric, where we denote by $\mathbf{v}$ and $\hat{\mathbf{v}}$ the reference and predicted velocities, respectively:

$$
\mathcal{E}(\mathbf{v}, \hat{\mathbf{v}}) = \frac{\| \mathbf{v} - \hat{\mathbf{v}} \|^2}{v_{\text{norm}}^2} + \gamma \left\| \frac{\mathbf{v}}{\epsilon + \| \mathbf{v} \|} - \frac{\hat{\mathbf{v}}}{\epsilon + \| \hat{\mathbf{v}} \|} \right\|^2 \tag{6}
$$

with hyperparameters $\gamma$ and $\epsilon \ll 1$, and where $v_{\text{norm}}$ is a reference velocity magnitude. The second term of the metric (6) allows to match the flow direction, even in the regions of small flow magnitude.

We generate training data through a FEM-based solver of (5), on a $100 \times 100$ triangular grid, accounting for nearly 91K degrees of freedom. To train LDNets, we take 100 evenly distributed snapshots in time, and we randomly take 200 points in space for each time step. We train three LDNets, by increasing the number of latent states from 1 to 5 and 10. The accuracy in the flow prediction for unseen inputs increases with the number of latent states (Fig. 4b and c). Furthermore, we challenge the trained LDNets in predicting the flow evolution even on a longer time horizon than that considered in the training dataset (specifically, twice as long). Remarkably, we observe a negligible propagation of the approximation error along the prolonged time frame, making the trained LDNets reliable also for time-extrapolation (Fig. 4b and d).

## Test Case 3: 1D electrophysiology model

We consider a nonlinear system of partial and ordinary differential equations describing the propagation of the electrical potential $z(x, t)$ in an excitable tissue, namely the Monodomain equation coupled with the Aliev-Panfilov (AP) model[52,58]. The AP model envisages a recovery variable $w(x, t)$ that tracks the refractoriness of the tissue by modulating the repolarization phase. The model, supplemented with homogeneous Neumann boundary conditions (encoding electrical insulation) and zero initial conditions for both the variables, reads

$$
\begin{aligned}
\frac{\partial z}{\partial t} - D \frac{\partial^2 z}{\partial x^2} = Kz(1-z)(z-\alpha) - zw + I_{\text{stim}}(x, t) & \quad x \in (0, L), t \in (0, T], \\
\frac{\partial w}{\partial t} = \left(\gamma + \frac{\mu_1 w}{\mu_2 + z}\right)(-w - Kz(z - b - 1)) & \quad x \in (0, L), t \in (0, T].
\end{aligned} \tag{7}
$$

The excitation-propagation process is triggered by an external stimulus $I_{\text{stim}}(x, t)$, applied at two stimulation points, respectively located at $x = 1/4L$ and $x = 3/4L$, and consisting of square impulses, to mimic the action of a (natural of artificial) pacemaker. The AP model solution features the fast-slow dynamics of a cardiac action potential (steep depolarization fronts followed by slow repolarization of the electrical potential to its resting value) and the wavefront propagation in space generating collisions of waves from different stimulation points. These features make this problem a challenging test case for comparing the proposed method against popular approaches to learning space-time dynamics of complex systems.

We compare LDNets with state-of-the-art approaches in which dimensionality reduction is achieved by training an auto-encoder (AE)

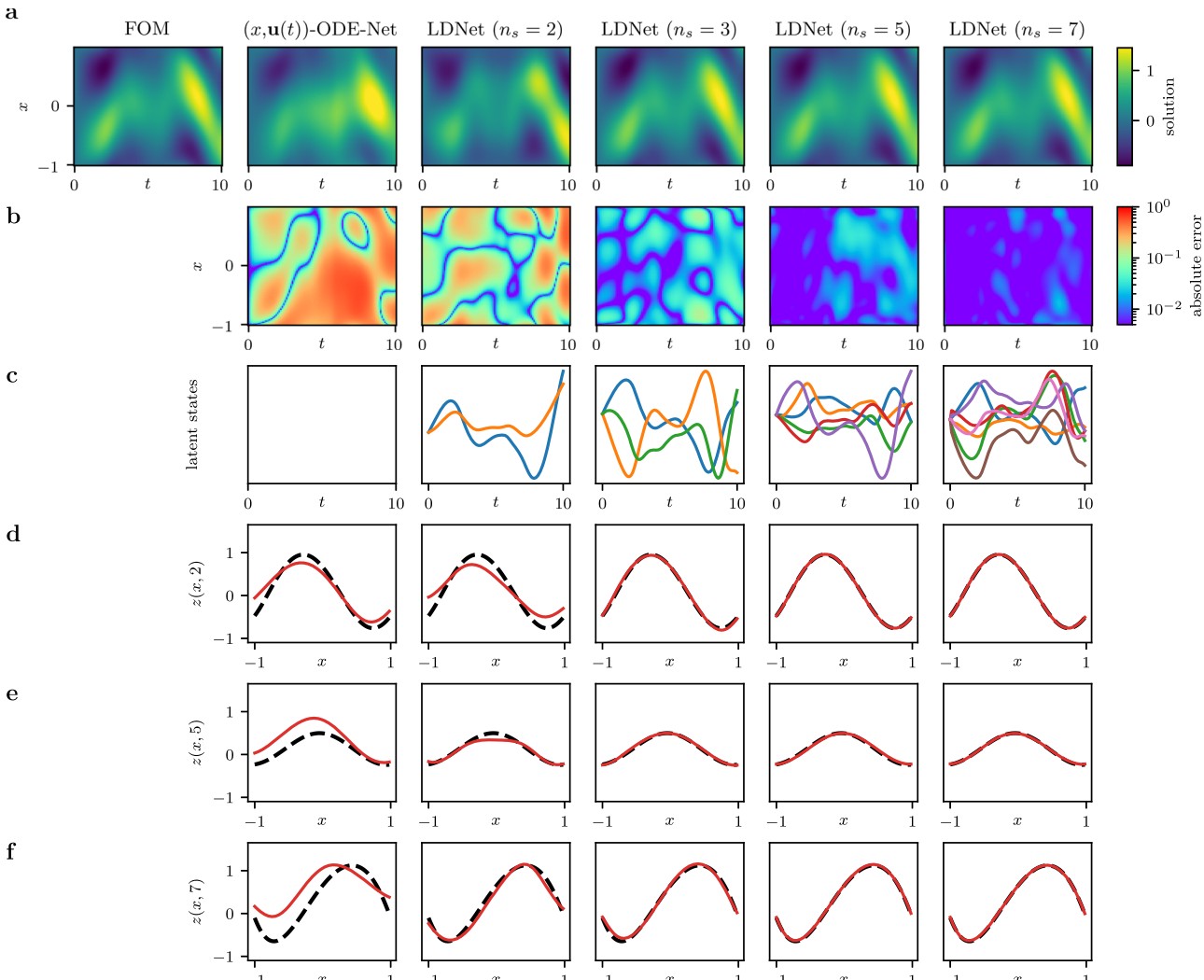

**Fig. 3 | Test Case 1c: impact of latent states.** We compare, for a sample belonging to the test dataset, the results obtained by using LDNets with increasing number of latent states (2, 3, 5, 7) and by using an ODE-Net fed by the input signal and the query point (denoted by $(\mathbf{x}, \mathbf{u}(t))$-ODE-Net). The left-most column reports the FOM solution (the abscissa denotes time, the ordinate denotes space). For each method we report: **a** the space-time solution; **b** the space-time error with respect to the FOM solution; **c** the time-evolution of the latent variables; **d**–**f** three snapshots of the space-dependent output field at $t = 2$, 5 and 7, in which we compare the predicted solution (red solid line) with the FOM solution (black dashed line).

on a discrete representation of the output $z(\cdot, t)$. Once trained, the encoder is employed to compute the trajectories of the latent states throughout the training set, and the dynamics in the latent space is learned either through an ODE-Net[32] or an LSTM[59]. We denote the resulting models by AE/ODE and AE/LSTM, respectively. Then, we further train the NN that tracks the dynamics of the latent states simultaneously with the decoder, that is in an end-to-end (e2e) fashion, and we denote the resulting models by AE/ODE-e2e and AE/LSTM-e2e, respectively. Furthermore, we benchmark LDNets against a classical method of model-order reduction of PDE models, namely the POD-DEIM method[60,61]. These methods are described in detail in the SI.

We challenge LDNets and the above-mentioned methods in the task of predicting the space-time dynamics of the target value $\mathbf{y}(x, t) = z(x,t)$, given the time series of impulses in the two stimulation points. To ensure a fair comparison, we rely on an automatic tuning algorithm to select the optimal hyperparameter values for the different methods, setting an upper bound of $d_s \leq 12$ on the latent space dimension. The reported results are obtained with the optimal hyperparameter configuration selected by the tuning algorithm, independently for each method.

The results of this comparison are reported in Figs. 5–6 and Table 1. Due to the presence of traveling fronts, this problem features a slow decay of the Kolmogorov $n$-width[7], that reflects in a poor accuracy of the electrical potential reconstruction given by the POD-DEIM method when 12 modes are used. As shown in the SI (see also Supplementary Movies 21–30), more than 24 modes are needed to achieve acceptable results, but this is accompanied by an increase of the computational cost in the prediction phase (see Table 1). A better accuracy is achieved by both auto-encoder-based methods and by LDNets, thanks to their ability to express a nonlinear relationship between the latent states and the solution. Still, LDNet outperforms the other methods, with a testing NRMSE equal to $7 \cdot 10^{-3}$. The testing NRMSE of auto-encoder-based methods is nearly 5 times larger than with LDNets or more. Remarkably, our method achieves better accuracy with significantly fewer trainable parameters: auto-encoder-based methods require more than tenfold the number of parameters. This testifies to the good architectural design of LDNets.

Furthermore, we observe that the POD-DEIM method results in a very limited speed-up with respect to the other methods considered. This limitation is intertwined with the necessity, due

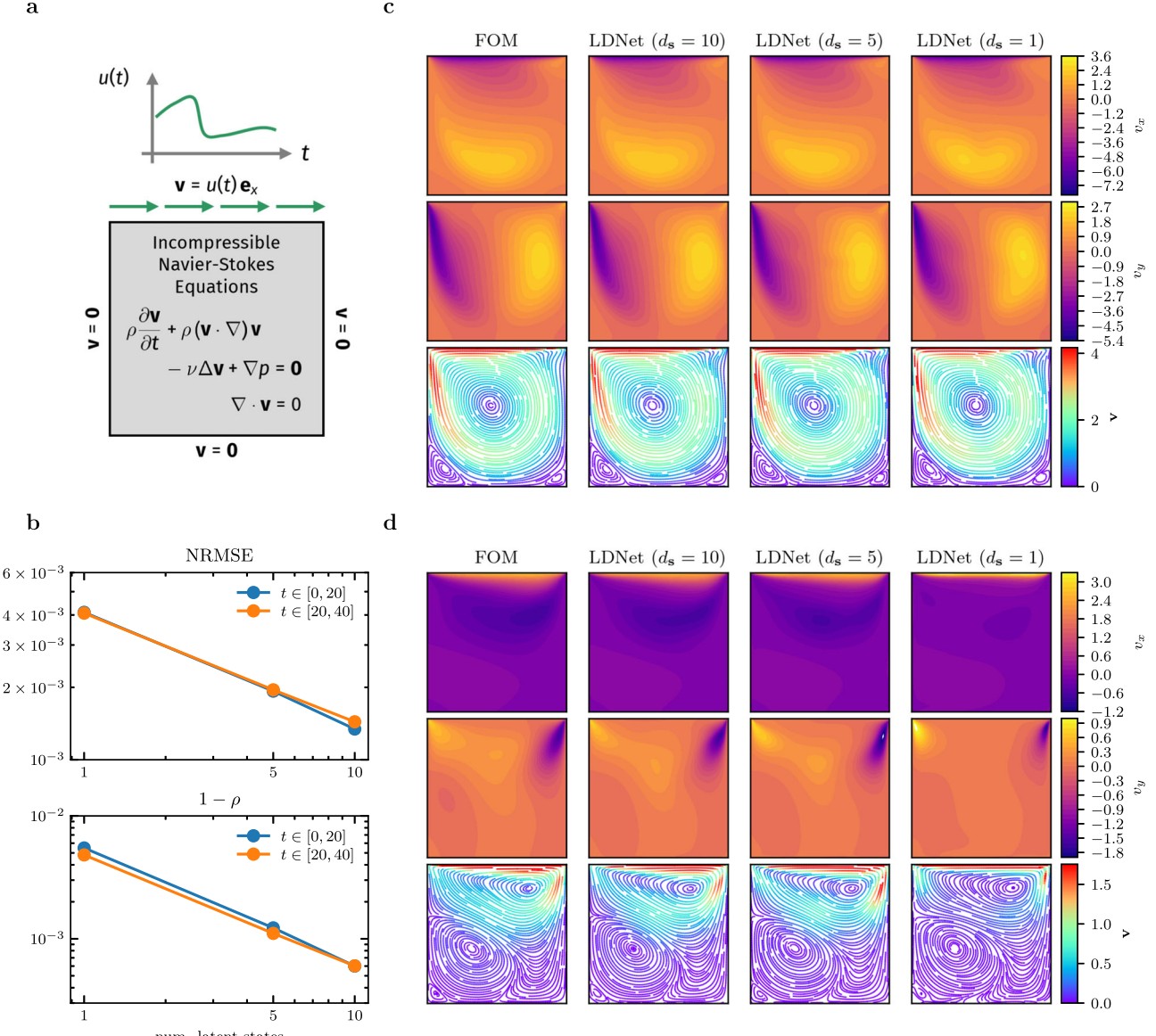

**Fig. 4 | Test Case 2. a** Computational domain and equations of the FOM. **b** Error metrics (NRMSE and Pearson dissimilarity) of LDNets for different number of latent variables ($d_s = 1$, 5 and 10). The training dataset consists of 80 simulations with $T = 20$, while the test dataset comprises 200 simulations with $T = 40$. The blue lines refer to the test error obtained in the interval $t \in [0, 20]$ (that is the same interval seen during training), while orange lines refer to the test error in the interval $t \in [0, 40]$. **c** A snapshot of the velocity field within the interval $t \in [0, 20]$ (interpolation interval) of a testing sample. **d** A snapshot of the velocity field within the interval $t \in [20, 40]$ (extrapolation interval) of a testing sample. For an animated version of this figure, see Supplementary Movies 1–10.

to numerical stability reasons, for the POD-DEIM model to be solved on the same temporal discretization as the high-fidelity model. This requirement represents a considerable constraint compared to the other methods outlined in this paper. As a matter of fact, as shown in Table 1, the computational cost for each sample with the FOM is about 37 s, the POD-DEIM method with 60 modes allows it to be reduced to about 8 s, when the other methods all lead to times less than 0.02 s. To this amount of time must be added the time required to evaluate the solution given the latent state variables, which, however, depends on the number of time steps and points at which this is required. For auto-encoder methods, the points at which this evaluation occurs are pre-established, taking $8.9 \cdot 10^{-7}$ s for each timestep. Conversely, LDNets, thanks to their mesh-less nature, offer the flexibility to evaluate at arbitrary locations, requiring $1.9 \cdot 10^{-7}$ s for each point in time and space. In this test case, should we want to

evaluate the solution at all time steps and training points, this would correspond to about $4.5 \cdot 10^{-4}$ s for auto-encoder-based methods and $9.5 \cdot 10^{-3}$ s for LDNets. That said, we observe that, with the exception of POD/DEIM, the inference times associated with the other methods are virtually negligible compared to the time required to evaluate the Full Order Model (FOM).

Concerning the offline time, associated with model construction, the training cost of LDNets (22,887 s) is lower than that of auto-encoder-based methods, except for AE/LSTM (11,009 s), which, however, yields a poor accuracy in the predictions. In fact, the accuracy achieved by AE/LSTM is matched by LDNets after just 1354 s of training. On the other hand, the accuracy levels of AE/ODE and AE/ODE-e2e are attained by LDNet after 6510 and 9090 s, respectively. The POD-DEIM method, as expected, is characterized by a less heavy offline phase, which, however, does not lead to a speed-up comparable to the other methods in evaluation.

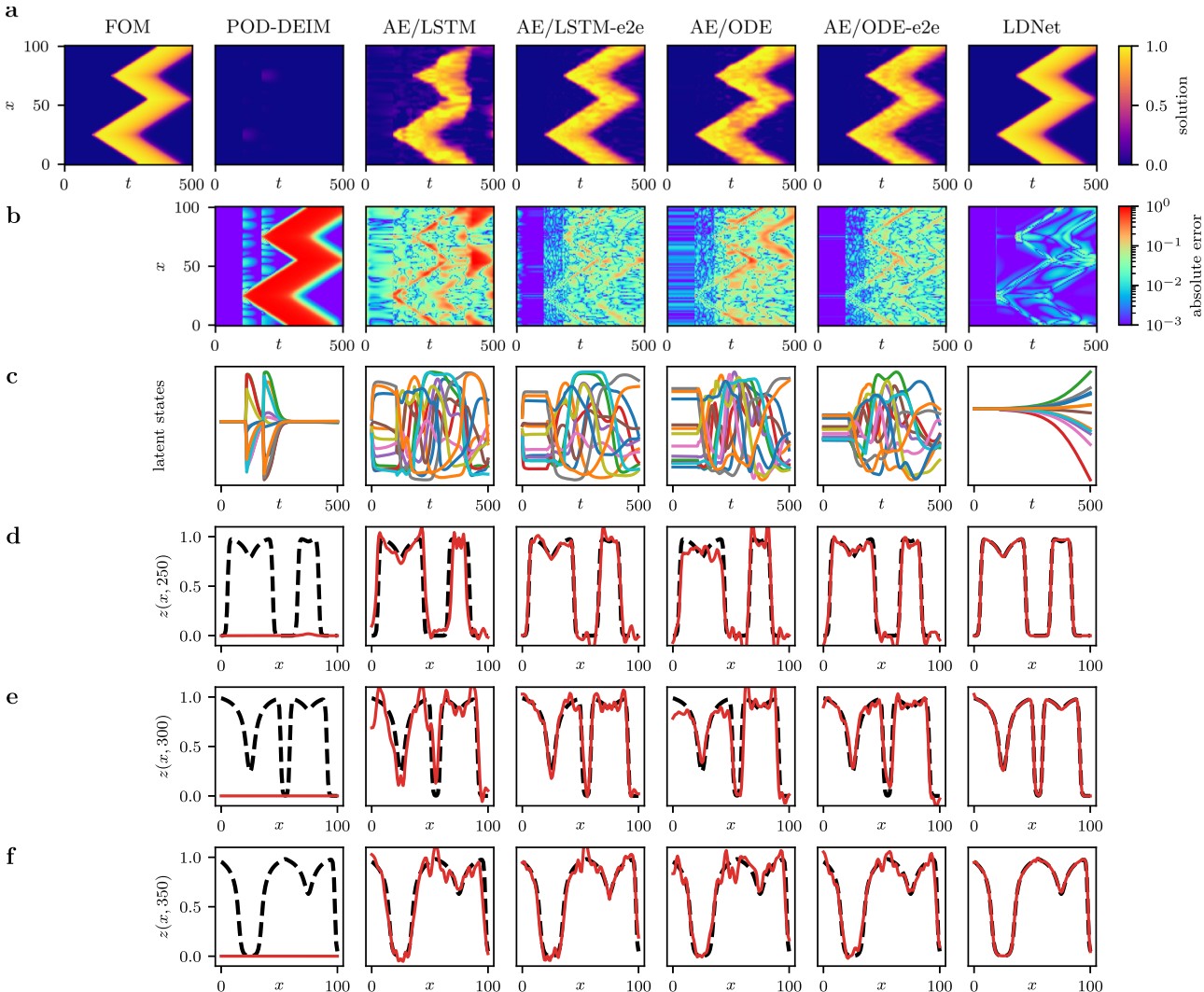

**Fig. 5 | Test Case 3: methods comparison.** We compare the results obtained with different methods for a sample belonging to the test dataset. The left-most column reports the FOM solution of the AP model (the abscissa denotes time, the ordinate denotes space). For each method we report: **a** the space-time solution; **b** the space-time error with respect to the FOM solution; **c** the time-evolution of the 12 latent variables; **d–f** three snapshots of the space-dependent output field at $t = 250$, 300 and 350, in which we compare the predicted solution (red solid line) with the FOM solution (black dashed line). For an animated version of this figure, see Supplementary Movies 11–20.

## Test Case 4: 2D electrophysiology model with reentrant activity

We consider the induction and sustainment of reentrant activity based on a two-dimensional version of the electrophysiological model (7). The experiment, inspired by[62], involves a first rightward propagating wavefront, followed by a second circular stimulus $I_{\text{stim}}(x, t)$ applied at the center of the square domain. Depending on the radius of the stimulus and on the stimulation time, three possible scenarios arise:

1. tissue refractoriness: the solution does not present a second activation because the circular stimulus is delivered while the tissue is still in a refractory state;
2. focal activation: the solution presents a single second focal activation originated by delivering the circular stimulus after the so-called vulnerable window;
3. reentrant drivers: the solution presents two self-sustained reentrant drivers that continuously reactivate the tissue.

Differently from Test Case 3, where we considered a one-dimensional wave propagation, here more complex spatial patterns with bifurcating phenomena are possible, as described above. We therefore want to test the ability of the proposed method in learning the spatio-temporal dynamics of this electrophysiological model,

upon variations of the stimulation radius and timing. Moreover, we further compare LDNets against state-of-the-art methods. For the sake of brevity, we only examine the three methods that have proven to perform best in Test Case 3, namely LDNet, AE/ODE and AE/ODE-e2e. Again, we select hyperparameters, independently for each algorithm, by means of the tuning algorithm described above, so as to ensure a fair comparison. The selected hyperparameters and further details on this test case are reported in the SI.

The comparison results are shown in Figs. 7–8 and Table 2. The results show that in this test case, which, compared to Test Case 3, has the added complexity of an extra spatial dimension, the advantage of LDNets over the considered methods is even more marked. As shown in Fig. 8, auto-encoder-based methods exhibit diverse artifacts in the solution, and, in particular, they fall short in accurately representing scenarios where tissue refractoriness does not result in signal propagation. The LDNet, on the other hand, produces predictions that are almost indistinguishable from those of the FOM, and is able to well capture the three different behaviors presented by the system considered in this test case. As a matter of fact, the LDNet achieves an RMSE on the test set that is more than 5.5 times smaller with respect to the other methods, despite using a significantly more parsimonious

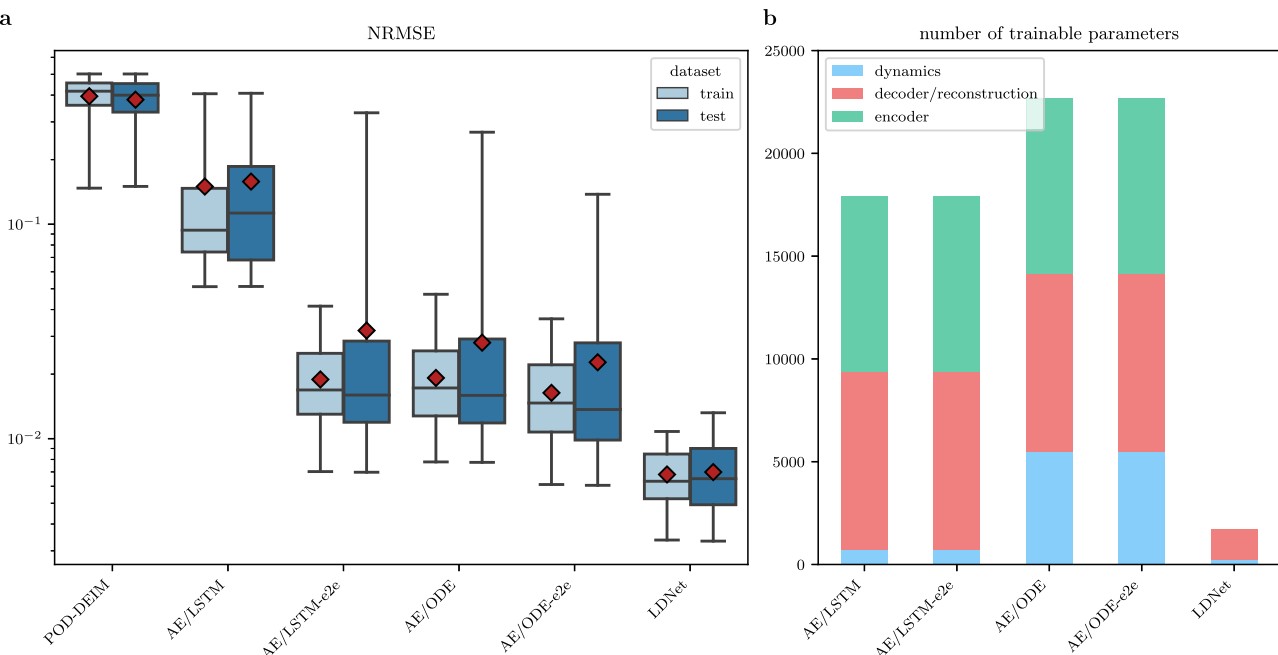

**Fig. 6 | Results of Test Case 3. a** Boxplots of the distribution of the testing (blue) and training (light blue) errors obtained with each method. The boxes show the quartiles while the whiskers extend to show the rest of the distribution. The red diamonds represent the average error on each dataset. **b** Number of trainable parameters of each method. The bin encoder is present only for auto-encoder-based methods, but not for LDNets. The bin dynamics refers to the NN that evolves the latent states. The POD-DEIM method is not included, as it does not envisage a training stage.

number of parameters (2.2 thousand, compared with more than 1 million for auto-encoder-based methods). The trained model inference time (online time) is comparable among the three methods considered. The time required to reconstruct the solution from the latent states for auto-encoder-based methods is $1.1 \cdot 10^{-4}$ s per time instant, while for LDNet it is $3.9 \cdot 10^{-6}$ s per time and space point. In all cases, the methods considered lead to a remarkable speedup with respect to the time required by the FOM (807 s per simulation). As for the offline stage, the time required to complete training with the three models is similar (nearly between 75,000 and 95,000 s). The LDNets, however, achieve higher levels of accuracy in less time. In fact, the accuracy achieved with AE/ODE is reached by LDNet after only 3569 s, while that of AE/ODE-e2e after 4530 s.

## Discussion

We have introduced LDNets, a class of NNs that learn in a data-driven manner the evolution of systems exhibiting spatio-temporal dynamics in response to external input signals.

An LDNet is trained in a supervised way from observations of input-output pairs, which can either come from experimental measurements or be synthetically generated through the numerical approximation of mathematical models of which one seeks a surrogate or reduced-order model. This latter case is the one considered in this manuscript to demonstrate the capabilities of the proposed method.

LDNets provide a paradigm-shift from state-of-the-art methods based on dimensionality reduction (e.g., exploiting POD or auto-encoders) of a high-dimensional discretization of the system state. Specifically, LDNets automatically discover a compact representation of the system state, without necessitating the explicit construction of an encoder. This enables the training algorithm to select a compact representation of the state that is functional not only in reconstructing the space-dependent field for each time instant, but also in predicting its dynamics; an auto-encoder, conversely, when trained, extracts features on a purely statistical basis, being agnostic of the importance of each feature in determining the evolution of the system. The latent states allow indeed the trained model to capture non-Markovian

effects by tracking the system history, and to propagate nonlocal information across the domain.

Unlike standard approaches that reconstruct a high-dimensional discretization of the output, corresponding e.g. to evaluations at the vertices of a computational mesh, our approach is in this sense meshless. The reconstruction NN is indeed queried for each point in space independently. This design principle gives LDNets several benefits. First, the meshless nature of LDNets combined with the automatic discovery of the latent space allows them to operate in a low-dimensional space without ever going through a high-dimensional discretization, as auto-encoder-based methods do. This makes LDNets very lightweight structures, easy to train, and not prone to overfitting. The LDNet architecture enables the sharing of the trainable parameters needed to evaluate the solution at different points (that is, the same weights are employed regardless of the query point). The low overfitting of LDNets is thus not surprising, as weight-sharing is often the key of good generalization properties of many architectures, such as CNNs and RNNs[49]. Second, it provides a continuous representation of the output, and, thus, allows for additional and possibly physics-informed terms to be introduced into the loss function[63], opening up countless possibilities for extending the purely black-box method proposed in this paper to grey-box approaches. Third, the loss function can be defined by stochastically varying the points in space at which the error is evaluated (see Test Case 2 and 4), thus lightening the computational burden associated with training. Note that this is not possible when the model returns the entire batch of observations. This aspect also opens up to multiple developments, such as stochastic, minibatch-based training algorithms, or even adaptive refinements of the evaluation points, by sampling more densely where the error is larger.

The time-dynamics of LDNets is based on a recurrent architecture that is consistent, by construction, with the arrow of time. This differentiates LDNets from other approaches in which time is seen as a parameter[30], or approaches, based e.g. on DeepONets, that take as input the entire time-history of $\mathbf{u}(t)$ with a fixed length[19,64]. The latter approaches do not easily allow for predictions over time frames longer

**Table 1 | Test Case 3: metrics of methods comparison**

| | NRMSE | | Number of trainable parameters | | | | Wall time (s) | |
|---|---|---|---|---|---|---|---|---|
| | training | testing | $\mathcal{NN}_{enc}$ | $\mathcal{NN}_{dec}, \mathcal{NN}_{rec}$ | $\mathcal{RNN}_{dyn}$ | total | offline | online |
| FOM | | | | | | | | 37.321 |
| POD-DEIM ($d_s = 12$) | $4.05 \cdot 10^{-1}$ | $3.92 \cdot 10^{-1}$ | | | | | 797 | 5.839 |
| POD-DEIM ($d_s = 24$) | $3.59 \cdot 10^{-1}$ | $3.47 \cdot 10^{-1}$ | | | | | 799 | 7.720 |
| POD-DEIM ($d_s = 36$) | $1.71 \cdot 10^{-1}$ | $1.62 \cdot 10^{-1}$ | | | | | 861 | 7.442 |
| POD-DEIM ($d_s = 48$) | $7.48 \cdot 10^{-2}$ | $7.57 \cdot 10^{-2}$ | | | | | 1124 | 7.976 |
| POD-DEIM ($d_s = 60$) | $2.97 \cdot 10^{-2}$ | $2.90 \cdot 10^{-2}$ | | | | | 1242 | 8.408 |
| AE/LSTM | $1.90 \cdot 10^{-1}$ | $1.98 \cdot 10^{-1}$ | 8562 | 8651 | 720 | 17,933 | 11,009 | 0.005 |
| AE/LSTM-e2e | $2.05 \cdot 10^{-2}$ | $5.87 \cdot 10^{-2}$ | 8562 | 8651 | 720 | 17,933 | 33,851 | 0.005 |
| AE/ODE | $2.09 \cdot 10^{-2}$ | $4.58 \cdot 10^{-2}$ | 8562 | 8651 | 5484 | 22,697 | 23,982 | 0.017 |
| AE/ODE-e2e | $1.78 \cdot 10^{-2}$ | $3.37 \cdot 10^{-2}$ | 8562 | 8651 | 5484 | 22,697 | 97,821 | 0.017 |
| LDNet | $7.09 \cdot 10^{-3}$ | $7.37 \cdot 10^{-3}$ | 0 | 1480 | 228 | 1708 | 22,887 | 0.014 |

Training and test errors obtained with the different methods, number of trainable parameters, and wall time associated with the offline phase and online phase. Computational times are obtained on a Intel Xeon Processor E5-2640 2.4 GHz. The offline phase refers to the construction of the model: for POD/DEIM, this involves building the basis for the solution manifold and for DEIM, while for the other methods it is associated with the NN training. The online phase, instead, involves predicting the evolution of the system for a new sample once the model has been constructed. This timeframe is referred to a single sample and excludes the evaluation of the output field, given its dependence on the number of considered time and space points. Further details are provided in the main text.

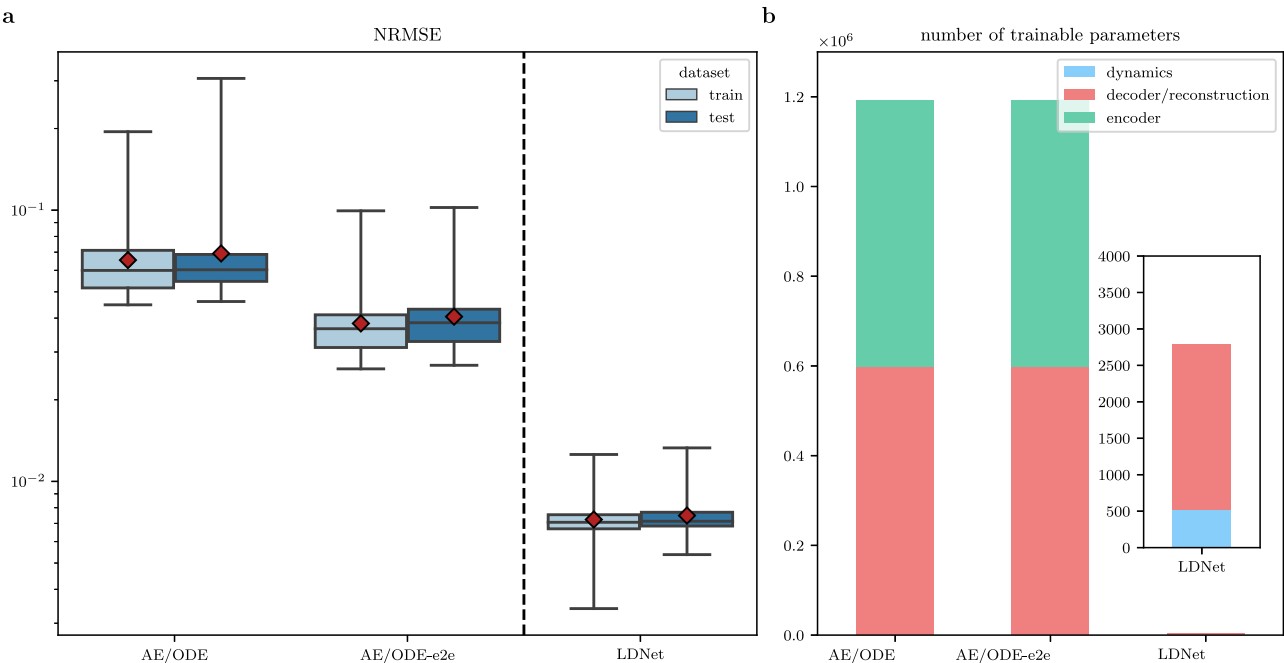

**Fig. 7 | Results of Test Case 4. a**: Boxplots of the distribution of the testing (blue) and training (light blue) errors obtained with different methods. The boxes show the quartiles while the whiskers extend to show the rest of the distribution. The red diamonds represent the average error on each dataset.

**b**: Number of trainable parameters of each method. The bin encoder is present only for auto-encoder-based methods, but not for LDNets. The bin dynamics refers to the NN that evolves the latent states. The inset shows an enlargement relative to LDNet.

than those used during training, or allow for time-extrapolation only in periodic o quasi-periodic problems[65]. LDNets, on the other hand, allow predictions for arbitrarily long times. We remark that the reliability of time-extrapolation is constrained by the characteristics of the problem at hand and the available training data. For example, if the system is characterized by a divergent behavior such that, as time progresses, the state enters regions increasingly distant from the initial condition, then the reliability of the predictions is not guaranteed in time-extrapolation regimes. When the system state remains bounded, however, the predictions of LDNets are significantly accurate even in time-extrapolation regimes, as showcased in Test Case 2.

We notice that the trajectories of the latent states $\mathbf{s}(t)$ obtained with LDNets are smoother than those obtained with auto-encoder-

based methods (see Fig. 5). This difference can be understood by considering how the latent state is constructed within auto-encoder-based methods. First, these methods learn a compact encoding of the high-dimensional output, thus defining a low-dimensional set of state variables, and then they attempt to find a law ruling their time evolution. However, while training the auto-encoder, the latent space is constructed with the sole purpose of allowing the output to be accurately reconstructed, without it necessarily being significant to the system dynamics. This issue is partially mitigated by a subsequent end-to-end training phase, which partially redefines the state variables in a way that is functional not only to reconstruct the solution, but also to capture the dynamics of the system. LDNets, instead, thanks to the simultaneous training of the dynamics NN and the reconstruction NN,

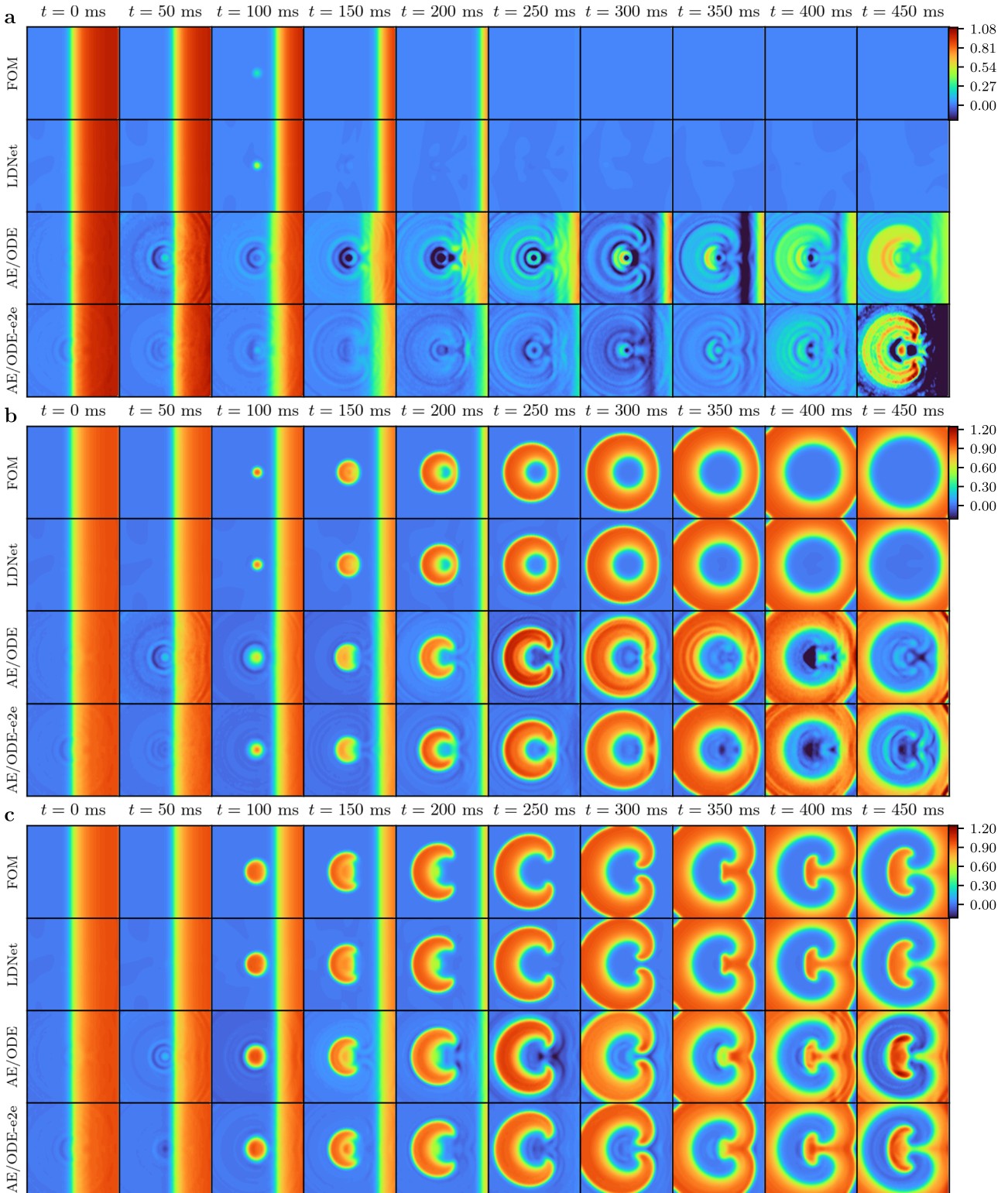

**Fig. 8 | Test Case 4: solution comparison.** Snapshots of the solution obtained with the different methods (reported on the left) at different time instants (reported on top). The figure refers to three samples belonging to the test set, corresponding to three different behaviors of the system: **a** tissue refractoriness; **b** focal activation; **c** reentrant drivers. For an animated version of this figure, see Supplementary Movie 31.

do not incur in this issue, as the training algorithm seeks the latent space that simultaneously pursue the twofold role of tracking the system dynamics and reconstructing the output at each time.

LDNets represent, as proved by the results of this work, an innovative tool capable of learning spatio-temporal dynamics with great accuracy and by using a remarkably small number of trainable parameters. They are able to discover, simultaneously with the system dynamics, compact representations of the system state, as shown in Test Case 1 where the Fourier transform of a sinusoidal signal is automatically discovered. Once trained, LDNets provide

**Table 2 | Test Case 4: metrics of methods comparison**

| | NRMSE | | Number of trainable parameters | | | | Wall time (s) | |
|---|---|---|---|---|---|---|---|---|
| | training | testing | $\mathcal{NN}_{\text{enc}}$ | $\mathcal{NN}_{\text{dec}}, \mathcal{NN}_{\text{rec}}$ | $\mathcal{RNN}_{\text{dyn}}$ | total | offline | online |
| FOM | | | | | | | | 807.210 |
| AE/ODE | $6.96 \cdot 10^{-2}$ | $7.83 \cdot 10^{-2}$ | 594,889 | 597,574 | 1269 | 1193,732 | 75,315 | 0.191 |
| AE/ODE-e2e | $3.97 \cdot 10^{-2}$ | $4.23 \cdot 10^{-2}$ | 594,889 | 597,574 | 1269 | 1193,732 | 95,479 | 0.188 |
| LDNet | $7.31 \cdot 10^{-3}$ | $7.57 \cdot 10^{-3}$ | 0 | 2276 | 513 | 2789 | 90,349 | 0.139 |

Training and test errors obtained with the different methods, number of trainable parameters, and wall time associated with the offline phase and online phase. See caption of Fig. 1 for more details.

predictions for unseen inputs with negligible computational effort (order of milliseconds for the considered Test Cases). LDNets provide a flexible and powerful tool for data-driven emulators that is open to a wide range of variations in the definition of the loss function (like, e.g., including physics-informed terms), in the training strategies, and, finally, in the NN architectures. The comparison with state-of-the-art methods on a challenging problem, such as predicting the excitation-propagation pattern of a biological tissue in response to external stimuli, highlights the full potential of LDNets, which outperform the accuracy of existing methods while still using a significantly lighter architecture.

A limitation of our work is that it does not consider the case of space-dependent inputs and of variable initial conditions, which will be the subject of future works; still, we remark that the class of problems that can be tackled with the proposed method encompass a broad range of real-life applications. Future developments will also focus on the topic of interpretability of the latent space, which is not covered in this work.

## Methods
### Notation
We denote input signals as $\mathbf{u}: [0, T] \to U$, taking values in the set $U \subseteq \mathbb{R}^{d_{\mathbf{u}}}$, and we denote by $\mathcal{U} \subseteq \{\mathbf{u} : [0, T] \to U\}$ the set of admissible input signals. Then, we denote by $\mathbf{y}: \Omega \times [0, T] \to Y$ the output (space-time dependent) field, with values in $Y \subseteq \mathbb{R}^{d_{\mathbf{y}}}$. For each time $t \in [0, T]$, the output field is defined within a space domain $\Omega \subset \mathbb{R}^d$. Finally, we denote by $\mathcal{Y} \subseteq \{\mathbf{y} : \Omega \times [0, T] \to Y\}$ the space of possible outputs. We assume that the map $\mathbf{u} \mapsto \mathbf{y}$ is well defined (i.e. the output $\mathbf{y}$ is unambiguously determined by the input $\mathbf{u}$) and it is consistent with the arrow of time (i.e. $\mathbf{y}(\mathbf{x}, t)$ depends on $\mathbf{u}(s)|_{s \in [0, t]}$ but not on $\mathbf{u}(s)|_{s \in (t, T)}$).

A relevant case is represented by a map $\mathbf{u} \mapsto \mathbf{y}$ defined as the composition of an observation operator and the solution map $\mathbf{u} \mapsto \mathbf{z}$ of a partial differential equation (PDE) in the form of (1), where $\mathbf{z} \in \mathcal{Z} \subseteq \{\mathbf{z} : \Omega \times [0, T] \to Z\}$, with $Z \subseteq \mathbb{R}^{d_{\mathbf{z}}}$, is the state variable (typically, $\mathcal{Z}$ is a Sobolev space). Here, $\mathbf{z}_0 : \Omega \to Z$ is the initial state, $\mathcal{F}$ is a differential operator, and $\mathcal{G}$ is the observation operator. We remark that, in this paper, neither knowledge nor even the existence of a model such as (1) is required: the training of an LDNet only requires input-output pairs.

**Remark 1.** The case when the output field $\mathbf{y}$ is determined not only by some time-dependent inputs $\mathbf{u}$, but also by some inputs that are time-independent (typically called parameters) is a special case of the one considered here. Still, to keep the notation compact, we use the same symbol $\mathbf{u}$ to collectively denote time-dependent inputs (i.e. signals) and time-constant inputs (i.e. parameters).

**Remark 2.** The full-order model (FOM) of (1) is an autonomous system. The non-autonomous case can be recovered as a special case by setting $\mathbf{u}(t) \cdot \mathbf{e}_k = t$ for some $k$, where $\mathbf{e}_k$ is the $k$th element of the canonical base of $\mathbb{R}^{d_{\mathbf{u}}}$.

### Training data
The training data are collected by considering a finite number of realizations of the map $\mathbf{u} \mapsto \mathbf{y}$, each one referred to as a training sample. For each training sample $i \in \mathcal{S}_{\text{train}}$, we collect the following discrete observations:
- $\mathbf{u}_i(\tau)$, for $\tau \in \mathcal{S}^i$;
- $\mathbf{y}_i(\boldsymbol{\xi}, \tau)$, for $\tau \in \mathcal{T}^i, \boldsymbol{\xi} \in \mathcal{P}_\tau^i$;

where $\mathcal{S}^i \subset [0, T]$, $\mathcal{T}^i \subset [0, T]$ and $\mathcal{P}_\tau^i \subset \Omega$ are discrete sets of observations. We remark that the observation times and points can be either shared among samples (i.e. $\mathcal{S}^i \equiv \mathcal{S}, \mathcal{T}^i \equiv \mathcal{T}$ and $\mathcal{P}_\tau^i \equiv \mathcal{P}$ for any $i$ and for any $\tau$) or be different from one sample to another.

Our goal is to learn the map $\mathbf{u} \mapsto \mathbf{y}$, that is to infer the output $\mathbf{y}(\mathbf{x}, t)$ corresponding to inputs $\mathbf{u}(t)$ outside the training set.

### LDNets
An LDNet is made of two fully-connected neural networks (FCNNs), namely the dynamics network $\mathcal{NN}_{\text{dyn}}$, with trainable parameters $\mathbf{w}_{\text{dyn}}$, and the reconstruction network $\mathcal{NN}_{\text{rec}}$, with trainable parameters $\mathbf{w}_{\text{rec}}$. The LDNet defines a map from a time-dependent input signal $\mathbf{u} \in \mathcal{U}$ to a space-time dependent field $\widetilde{\mathbf{y}} \in \mathcal{Y}$ through the solution of the following system of ordinary differential equations (ODEs):

$$\begin{cases} \dot{\mathbf{s}}(t) = \mathcal{NN}_{\text{dyn}}(\mathbf{s}(t), \mathbf{u}(t); \mathbf{w}_{\text{dyn}}) & \text{in } (0, T] \\ \mathbf{s}(0) = \mathbf{0} \\ \widetilde{\mathbf{y}}(\mathbf{x}, t) = \mathcal{NN}_{\text{rec}}(\mathbf{s}(t), \mathbf{u}(t), \mathbf{x}; \mathbf{w}_{\text{rec}}) & \text{for } \mathbf{x} \in \Omega \text{ and } t \in [0, T], \end{cases} \quad (8)$$

where $\mathbf{s}(t) \in \mathbb{R}^{d_{\mathbf{s}}}$ is the vector of latent states. The number of latent states $d_{\mathbf{s}}$ is set by the user, and should be regarded as an hyperparameter. The latent variables $\mathbf{s}(t)$ allow to keep track of the state of the system. These, however, are not defined a priori (unlike methods based on dimensionality reduction techniques, see SI), but the latent space is discovered during the training process. This is similar to[47] for the case of time signals as outputs and to the Recurrent Neural Operator (RNO)[22,23], used to learn microscopic internal variables capable of tracking the history dependence in multiscale materials. However, while in the RNO the latent variables correspond to a local material memory, LDNets have a single set of latent variables for the entire domain. In this work, we always consider hyperbolic tangent (tanh) activation functions.

**Remark 3.** The formulation (8) is the most general one. A special case is the one where $\mathcal{NN}_{\text{rec}}$ does not depend on $\mathbf{u}(t)$. Whether or not to include the latter dependency in $\mathcal{NN}_{\text{rec}}$ is an architectural choice that shall be regarded as a hyperparameter, possibly subject to selection via cross-validation. In many cases, however, the choice can be driven by the physics of the underlying process. Specifically, we will leave an explicit dependency whenever the output $\mathbf{y}(\mathbf{x}, t)$ depends on the input $\mathbf{u}(t)$ instantaneously. The case where the dependency is neglected is the one that we mostly consider in our test cases, expect for Test Case 2, in which we allow $\mathcal{NN}_{\text{rec}}$ to depend on $\mathbf{u}(t)$ in a direct way.

In practice, the ODE system (8) is discretized by a suitable numerical method. In this work, we employ a Forward Euler scheme with a uniform time step size $\Delta t$, but other schemes could be considered as well (e.g. time-adaptive Runge-Kutta schemes[53]). In case the observation times $\mathcal{S}^i$ do not coincide with the discrete times $k\Delta t$, for $k = 1, \ldots$, we perform a re-sampling of $\mathbf{u}$ through a piecewise linear interpolation. Similarly, to evaluate the predicted output $\widetilde{\mathbf{y}}$ in correspondence of the observation times $\tau \in \mathcal{T}^i$, we interpolate the discrete solution of $\mathbf{s}(t)$ at the time instants $\tau$.

We denote with the symbol $\mathcal{RNN}_{\mathrm{dyn}}$ (to evoke its recurrent nature) the operator mapping the time series of inputs $\{\mathbf{u}_i(\tau)\}_{\tau \in \mathcal{S}^i}$ associated with a given sample $i$ to the latent state $\mathbf{s}_i$ evolution. More precisely, we have, for any sample $i$ and at any time $t \in [0, T]$:

$$\mathbf{s}_i(t) = \mathcal{RNN}_{\mathrm{dyn}}(\{\mathbf{u}_i(\tau)\}_{\tau \in \mathcal{S}^i}, t; \mathbf{w}_{\mathrm{dyn}})$$

With this notation, the LDNet output $\widetilde{\mathbf{y}}_i(\mathbf{x}, t)$ is the result of the composition of $\mathcal{NN}_{\mathrm{rec}}$ with $\mathcal{RNN}_{\mathrm{dyn}}$:

$$\widetilde{\mathbf{y}}_i(\mathbf{x}, t) = \mathcal{NN}_{\mathrm{rec}}(\mathcal{RNN}_{\mathrm{dyn}}(\{\mathbf{u}_i(\tau)\}_{\tau \in \mathcal{S}^i}, t; \mathbf{w}_{\mathrm{dyn}}), \mathbf{u}_i(t), \mathbf{x}; \mathbf{w}_{\mathrm{rec}}). \quad (9)$$

To train the LDNet, we define the loss function:

$$\mathcal{L}(\mathbf{w}_{\mathrm{dyn}}, \mathbf{w}_{\mathrm{rec}}) = \underset{i \in \mathcal{S}_{\mathrm{train}}}{\sum} \underset{\tau \in \mathcal{T}^i}{\sum} \underset{\boldsymbol{\xi} \in \mathcal{P}^i_\tau}{\sum} \mathcal{E}(\widetilde{\mathbf{y}}_i(\boldsymbol{\xi}, \tau), \mathbf{y}_i(\boldsymbol{\xi}, \tau))$$
$$+ \alpha_{\mathrm{dyn}} \mathcal{R}(\mathbf{w}_{\mathrm{dyn}}) + \alpha_{\mathrm{rec}} \mathcal{R}(\mathbf{w}_{\mathrm{rec}}),$$

where the symbol $\sum$ denotes the average operator (that is the sum over a set divided by the cardinality of the set), and where $\widetilde{\mathbf{y}}_i(\boldsymbol{\xi}, \tau)$ are the outputs of the LDNet associated with the trainable parameters $\mathbf{w}_{\mathrm{dyn}}$ and $\mathbf{w}_{\mathrm{rec}}$ as defined in (9). The discrepancy metric $\mathcal{E}$ is typically defined as

$$\mathcal{E}(\widetilde{\mathbf{y}}, \mathbf{y}) = \frac{\|\widetilde{\mathbf{y}} - \mathbf{y}\|^2}{y^2_{\mathrm{norm}}} \quad (10)$$

with $y_{\mathrm{norm}}$ being a normalization factor defined from case to case and where $\|\cdot\|$ denotes the euclidean norm. The first term of $\mathcal{L}$ represents therefore the normalized mean square error between observations and LDNet predictions. Moreover, to mitigate overfitting, suitable regularization terms on the NN weights could be introduced, with weighting factors $\alpha_{\mathrm{dyn}}$ and $\alpha_{\mathrm{rec}}$. In this work, we define $\mathcal{R}$ as the mean of the squares of the NN weights (yielding the so-called $L^2$-regularization or Tikhonov regularization).

**Remark 4.** The quadratic discrepancy metric (10), while being the most natural choice, is not the unique one. For instance, it can be replaced by goal-oriented metrics (an example is given in Test Case 2).

Training an LDNet consists in employing suitable optimization methods to approximate the solution of the following non-convex minimization problem:

$$(\mathbf{w}^*_{\mathrm{dyn}}, \mathbf{w}^*_{\mathrm{rec}}) = \underset{\mathbf{w}_{\mathrm{dyn}}, \mathbf{w}_{\mathrm{rec}}}{\mathrm{argmin}} \mathcal{L}(\mathbf{w}_{\mathrm{dyn}}, \mathbf{w}_{\mathrm{rec}}).$$

The two NNs are simultaneously trained.

## Normalization layers

In order to facilitate training, we normalize the inputs and the outputs of the NNs. Specifically:

- We normalize the signals $\mathbf{u}$, the output fields $\mathbf{y}$ and the space variables $\mathbf{x}$, so that each entry approximately spans the interval $[-1, 1]$. We normalize each entry independently of the others. More precisely we normalize each scalar variable $\alpha$ through the

affine transformation $\tilde{\alpha} = (\alpha - \alpha_0)/\alpha_w$ where $\alpha_0$ is a reference value and $\alpha_w$ is a reference width. To define $\alpha_0$ and $\alpha_w$, we follow two different strategies.

1. If the variable takes values in a bounded interval $[\alpha_{\mathrm{min}}, \alpha_{\mathrm{max}}]$, we set

$$\alpha_0 = (\alpha_{\mathrm{min}} + \alpha_{\mathrm{max}})/2,$$
$$\alpha_w = (\alpha_{\mathrm{max}} - \alpha_{\mathrm{min}})/2.$$

2. If the variable is sampled from a distribution with unbounded support (e.g., when $\alpha$ is normally distributed), we set $\alpha_0$ equal to the sample mean and $\alpha_w$ equal to three times the sample standard deviation.

- We also normalize the time variable, by dividing the time steps by a characteristic time scale $\Delta t_{\mathrm{ref}}$. The normalization constant $\Delta t_{\mathrm{ref}}$ impacts the output of $\mathcal{NN}_{\mathrm{dyn}}$, that is dimensionally proportional to the inverse of time. Since finding a good value for $\Delta t_{\mathrm{ref}}$ is in general not straightforward, we typically consider it as a hyperparameter, tuned through a suitable automatic algorithm (described below).

- We do not normalize the latent states $\mathbf{s}$, since their distribution is not known before training. Indeed, when hyperparameters are well tuned, the training algorithm tends to generate models that produce latent states with approximately normalized values.

In practice, normalization can be achieved either by modifying the training data accordingly, or by embedding the two NNs between two normalization layers (namely, one input layer and one output layer) each. Formally, the second approach consists in defining $\mathcal{NN}_{\mathrm{dyn}}$ and $\mathcal{NN}_{\mathrm{rec}}$ as follows, where we $\widetilde{\mathcal{NN}}_{\mathrm{dyn}}$ and $\widetilde{\mathcal{NN}}_{\mathrm{rec}}$ are two FCNNs:

$$\mathcal{NN}_{\mathrm{dyn}}(\mathbf{s}, \mathbf{u}; \mathbf{w}_{\mathrm{dyn}}) = \Delta t^{-1}_{\mathrm{ref}} \widetilde{\mathcal{NN}}_{\mathrm{dyn}}(\mathbf{s}, (\mathbf{u} - \mathbf{u}_0) \oslash \mathbf{u}_w; \mathbf{w}_{\mathrm{dyn}})$$
$$\mathcal{NN}_{\mathrm{rec}}(\mathbf{s}, \mathbf{u}, \mathbf{x}; \mathbf{w}_{\mathrm{rec}}) = \mathbf{y}_0 + \mathbf{y}_w \odot \widetilde{\mathcal{NN}}_{\mathrm{rec}}(\mathbf{s}, (\mathbf{u} - \mathbf{u}_0) \oslash \mathbf{u}_w, (\mathbf{x} - \mathbf{x}_0) \oslash \mathbf{x}_w; \mathbf{w}_{\mathrm{rec}})$$

where $\odot$ and $\oslash$ denote the Hadamard (i.e. element-wise) product and division, respectively.

In case the distribution of a given output field features long tails, we introduce a nonlinear layer aimed at compressing them. The layer applies the transformation $y \mapsto (y^3 + \beta y)/(1 + \beta)$, where the hyperparameter $\beta > 0$ tunes the compression strength.

## Imposing a-priori physical knowledge

The architecture of LDNets reflects certain features of the physics they are meant to capture. With respect to the space variable, the representation is continuous, unlike methods that reconstruct a discretized solution thus losing the correspondence between neighboring points. With respect to the time variable, the dynamics is driven by a system of differential equations which makes LDNets consistent with the arrow of time (i.e., with the causality principle[47]). These features make it natural to introduce a-priori physical knowledge in the construction and training of LDNets. In this regard, we distinguish between weak imposition and strong imposition.

Weak imposition consists of introducing physics-informed terms[63] into the loss function, aimed at promoting solutions that satisfy certain requirements (such as irrotationality of a velocity field, to make an example). In this paper we do not show examples in this regard, but simply highlight that the continuous representation of the output field used by LDNets makes the introduction of such terms very straightforward through the use of automatic differentiation.

Strong imposition, on the other hand, consists of modifying the architecture of the LDNet components in order to obtain models that automatically satisfy certain properties[66,67]. In what follows, we provide two examples of how this can be applied to ensure both temporal (acting on $\mathcal{NN}_{\mathrm{dyn}}$) and spatial (acting on $\mathcal{NN}_{\mathrm{rec}}$) properties.

**Equilibrium configuration imposition.** In many real-life applications, data are collected starting from an equilibrium configuration. This entails that the initial state should be an equilibrium for the latent dynamics as well, in virtue of the interpretation of $\mathbf{s}$ as a compact encoding of the full-order system state. Therefore, we define the right-hand side of the latent state evolution equation as follows, where $\widetilde{\mathcal{NN}}_{\mathrm{dyn}}$ is a trainable FCNN and where $\mathbf{u}_{\mathrm{eq}} \in U$ is the input at equilibrium:

$$\mathcal{NN}_{\mathrm{dyn}}(\mathbf{s}, \mathbf{u}; \mathbf{w}_{\mathrm{dyn}}) = \widetilde{\mathcal{NN}}_{\mathrm{dyn}}(\mathbf{s}, \mathbf{u}; \mathbf{w}_{\mathrm{dyn}}) - \widetilde{\mathcal{NN}}_{\mathrm{dyn}}(\mathbf{0}, \mathbf{u}_{\mathrm{eq}}; \mathbf{w}_{\mathrm{dyn}})$$

As a consequence, the initial state $\mathbf{s} = \mathbf{0}$ of the model is an equilibrium for any choice of the trainable parameters $\mathbf{w}_{\mathrm{dyn}}$.

**Prescribed solution in subsets of the domain (e.g. Dirichlet boundary conditions).** The evolution of the output field is often unknown except on a subset of the domain $\Omega$, such as for example a portion $\Gamma_D$ of its boundary $\partial\Omega$. This happens, e.g., when there is a FOM that features a Dirichlet boundary condition like

$$\mathbf{y}(\mathbf{x}, t) = \mathbf{y}_D(\mathbf{x}) \quad \text{on } \Gamma_D. \tag{11}$$

In this case, the solution is constrained to satisfy (11) by defining $\mathcal{NN}_{\mathrm{rec}}$ as

$$\mathcal{NN}_{\mathrm{rec}}(\mathbf{s}, \mathbf{u}, \mathbf{x}; \mathbf{w}_{\mathrm{rec}}) = \mathbf{y}_{\mathrm{lift}}(\mathbf{x}) + \widetilde{\mathcal{NN}}_{\mathrm{rec}}(\mathbf{s}, \mathbf{u}, \mathbf{x}; \mathbf{w}_{\mathrm{rec}})\psi(\mathbf{x}),$$

where $\widetilde{\mathcal{NN}}_{\mathrm{rec}}$ is a trainable FCNN, $\mathbf{y}_{\mathrm{lift}}$ is the lifting of the boundary datum, that is an extension of $\mathbf{y}_D$ to the whole domain $\Omega$, and $\psi : \Omega \to \mathbb{R}$ is a mask, that is a smooth function such that $\psi(\mathbf{x}) = 0$ if and only if $\mathbf{x} \in \Gamma_D$. See[57] for further details and[68] for a general approach to construct the mask $\psi$ based on approximate distance functions.

### Training algorithm
To train the LDNet, we employ a two stage strategy. First, we perform a limited number of epochs (typically, a few hundreds) with the Adam optimizer[48], starting with a learning rate of $10^{-2}$. Then, we switch to a second-order accurate optimizer, namely BFGS[49].

To evaluate the gradient of the loss function with respect to the trainable parameters, we combine back-propagation-through-time for $\mathcal{RNN}_{\mathrm{dyn}}$ with back-propagation for $\mathcal{NN}_{\mathrm{rec}}$[49]. To initialize the parameters of the two NNs, we employ a Glorot uniform strategy for weights and zero values for the biases[49].

Training ODE-Nets often presents challenges and typically involves an adaptive time integration to deal with stiff dynamics, which makes the computational graph potentially very deep and the computational cost often prohibitive[69–72]. In this work, instead, we rely on a fixed time step size to integrate the latent variables. Thanks to the fact that the latent variables are not fixed a priori, but are defined at training stage, the training algorithm tends to define latent variables with non-stiff dynamics, whose evolution is well captured through a fixed time step size, regardless of the stiffness of full-order model employed to generate the data. An evidence for this is provided by Test Case 3: the ground-truth model (Aliev-Panfilov, Eq. (7)) features, as it is well known, very stiff dynamics[58], thus imposing the use of a timestep of $5 \cdot 10^{-6}$ s, whereas the LDNet succeeds in fitting the results with great accuracy while using a much larger timestep (equal to $1 \cdot 10^{-3}$ s). This behavior is observed in our preliminary work[73] as well.

### Hyperparameter tuning algorithms
The hyperparameters of the proposed method are the number of layers and neurons of $\mathcal{NN}_{\mathrm{dyn}}$ and $\mathcal{NN}_{\mathrm{rec}}$, the $L^2$ regularization weights $\alpha_{\mathrm{dyn}}$ and $\alpha_{\mathrm{rec}}$, the normalization time constant $\Delta t_{\mathrm{ref}}$ and, whenever necessary in the different test cases, the number of latent states $d_{\mathbf{s}}$. To

automatically tune them, we employ the Tree-structured Parzen Estimator (TPE) Bayesian algorithm[50,74]. The hyperparameters search space is defined as an hypercube, with a log-uniform sampling. We perform K-fold cross validation while monitoring the value of the discrepancy metric in Eq. (10). We also employ the Asynchronous Successive Halving (ASHA) scheduler to early terminate hyperparameters configurations that are either bad or not promising[51,75].

We simultaneously train multiple NNs associated to different hyperparameters settings on a supercomputer endowed with several CPUs via Message Passing Interface (MPI). Each NN exploits Open Multi-Processing (OpenMP) for Hyper-Threading, which allows for a speed-up in the computationally-intensive tensor operations involved during the training phase. For the implementation, we rely on the Ray Python distributed framework[76].

## Data availability
The data necessary to reproduce the results presented here are publicly available on Zenodo in the repository[77], available at the URL https://doi.org/10.5281/zenodo.10436489.

## Code availability
The software implementation of the proposed methodology and the code supporting the results presented here are publicly available in the `LDNets` repository at https://github.com/FrancescoRegazzoni/LDNets.

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

## Acknowledgements
F.R., S.P. and L.D. are members of the INdAM research group GNCS. This project has been partially supported by the INdAM GNCS Project CUP_E55F22000270001. The present research is part of the activities of "Dipartimento di Eccellenza 2023-2027", Department of Mathematics, Politecnico di Milano. L.D. acknowledges the support of the FAIR (Future Artificial Intelligence Research) project, funded by the NextGenerationEU program (Italy) within the PNRR-PE-AI scheme (M4C2, investment 1.3, line on Artificial Intelligence).

## Author contributions
F.R. conceived the methodology and developed its software implementation. F.R., S.P., and M.S. implemented the test cases and performed the analysis. F.R., S.P., and M.S. wrote the the manuscript. L.D. and A.Q. reviewed the manuscript and supervised the work.

## Competing interests
The authors declare no competing interests.
