## [Peer Review File · Nature Communications]

REVIEWER COMMENTS

Reviewer #1 (Remarks to the Author):

The manuscript introduces a LD-net architecture, which is specifically designed to learn the spatial-temporal solutions of PDEs governed by first-order dynamics. This network employs two FCNs, a dynamic net and a reconstruction network, which function to map the input function to latent variables and subsequently map these latent variables to the comprehensive solution of the PDE. The proposed method's efficacy is then tested on the Advection-Diffusion-Reaction equation, Unsteady Navier-Stokes, and the Aliev-Panfilov electrophysiology model.

I found the paper to be a compelling read, well-articulated and furnished with clear and easy-to-follow mathematical problem/method definitions. However, my principal concern pertains to the novelty of the work. The LD-net method, proposed by the paper, is essentially the same to the Recurrent Neural Operator [1,2], that is developed for the study of homogenization problems. Although the contexts of application differ considerably, the underlying structure and key conclusions appear to be fundamentally similar.

This critique does not intend to undermine the present work, given that its application differs substantially from the preceding studies, and it's intriguing to observe a similar structure's effective application across diverse problems. Nevertheless, I believe it is essential for the authors to acknowledge the Recurrent Neural Operator, pointing out similarities and differences since this architecture is not novel to their work.

There are also some minor issues I'd like to raise:

It would be beneficial for the authors to consider moving portions of the LD section currently in the appendix to the main text, as it contains vital information on how the LD-net is constructed and trained.

Test cases 1 and 2 indicate that the test error decreases as the number of latent variables increases. It would be interesting to know what happens when an excessive number of latent variables, say more than 5, are used. Will the error eventually converge to a constant, as is suggested by studies [1,2]?

Studies [1,2] have also indicated that the latent dynamics that can be discovered from the data are constrained by the training dataset. Thus, critical latent dynamics can be missed if the training data do not encompass this information. Does the same hold true for the PDEs examined here?

In conclusion, I believe this paper has considerable potential. However, it requires a thorough review and subsequent refinement by the authors. Given my observations, I recommend a major revision before further consideration.

References:

[1] Bhattacharya, K., Liu, B., Stuart, A., & Trautner, M. (2023). Learning Markovian homogenized models in viscoelasticity. *Multiscale Modeling & Simulation*, 21(2), 641-679.

[2] Liu, B., Ocegueda, E., Trautner, M., Stuart, A. M., & Bhattacharya, K. (2023). Learning macroscopic internal variables and history dependence from microscopic models. *Journal of the Mechanics and Physics of Solids*, 105329.

Reviewer #2 (Remarks to the Author):

The authors propose a network that utilizes 1) a neural ODE to propagate reduced order latent dynamics, and 2) a second network to map the low order dynamics to the high dimensional space.

The approach is mesh-less, avoiding the use of autoencoders, as the grid-point to be evaluated is passed as an input to the network. As the proposed networks do not operate on the high dimensional space, they are faster compared to other alternatives in the literature.

The authors test their method on the advection diffusion reaction equation, where they demonstrate that LDNets can capture the dominant low-order Fourier modes, and improve its accuracy

as the latent space is increased, in trivial test cases. LDNets are also benchmarked in the 2D

lid-driven cavity flow, where the task is to predict the velocity field from the time-varying forcing.

Last but not least, the LDNets are benchmarked to other standard approaches (autoencoders,

LSTMs, etc.) in the Aliev-Panfilov (AP) model, which is a PDE describing the propagation of the electrical potential in an excitable tissue, demonstrating lower error.

The language of the paper is clear. The analysis is scientifically sound. In general the claims / conclusions of the paper are supported by the analysis.

However, there are some methodological concerns.

1 Comments

- The novelty of the approach is the introduction of a reduced order latent space in the neural ODE, as all other main features (mesh-less, Neural ODEs, external forcing dependent

network) of the method can be found in various other works (PINNs, etc.). How does the

method compare to just using a Neural ODE with inputs the external forcing and the grid

point? Why does there have to be a latent space? The latent space is used by other methods for dimensionality reduction. It does not seem to be needed in this architecture, as the

gridpoint is provided in the input.

- In general the authors demonstrate that indeed LDNets can learn low order dynamics driven by an external forcing. However, the LDNets do not operate on the high dimensional space.

While this characteristic is viewed as a strength of the method, it appears unlikely that the

LDNets can adapt to scenarios where dynamics are driven by the initial condition or complex spatial correlations, rather than the forcing term, unless some modifications are incorporated.

This is an important pitfall of the proposed approach.

- The errors that the methods (LDNets, AE+LSTMs, etc.) demonstrate in included benchmark (Aliev-Panfilov model) are very small, close to machine precision. From a machine

learning standpoint, the task appears to be effectively solved by all methods. I am not sure

about the relevance of demonstrating that the proposed approach reaches errors closer to machine precision compared to other works.

- In Figure 5 for example, all models fairly low errors. The differences most likely stem from architectural choices, and hyper-parameter optimization.

- In Figure 4, it seems that the LDNets learn a latent space qualitatively very different compared to all other approaches. The latent space seems to be diverging exponentially. What

is the intuition behind that? Is that a desirable characteristic?

- The latent dynamics are propagated through a neural ODE. These networks are notoriously hard to train, and many works attempt to alleviate the associated problems [1, 2, 3, 4]. Did the authors employ any mechanism to alleviate the training problems?
- The authors do not provide results concerning the training time, and inference time (to represent the whole spatial field in testing) and comparisons with other methods.
- Publishing the code and data would assist reproducibility and strengthen the claims of the paper.

References

- [1] Xuanqing Liu, Tesi Xiao, Si Si, Qin Cao, Sanjiv Kumar, and Cho-Jui Hsieh. Neural sde: Stabilizing neural ode networks with stochastic noise. arXiv preprint arXiv:1906.02355, 2019.
- [2] Chris Finlay, Jörn-Henrik Jacobsen, Levon Nurbekyan, and Adam Oberman. How to train your neural ode: the world of jacobian and kinetic regularization. In International conference on machine learning, pages 3154–3164. PMLR, 2020.
- [3] Marin Biloš, Johanna Sommer, Syama Sundar Rangapuram, Tim Januschowski, and Stephan Günnemann. Neural flows: Efficient alternative to neural odes. *Advances in neural information processing systems*, 34:21325–21337, 2021.
- [4] Arnab Ghosh, Harkirat Behl, Emilien Dupont, Philip Torr, and Vinay Namboodiri. Steer: Simple temporal regularization for neural ode. *Advances in Neural Information Processing Systems*, 33:14831–14843, 2020.

November 21st, 2023

F. Regazzoni, S. Pagani, M. Salvador, L. Dede', A. Quarteroni

Learning the intrinsic dynamics of spatio-temporal processes through Latent Dynamics Networks

NCOMMS-23-26387-T

Answers to Reviewers

We have addressed all the issues raised by the Reviewers, who helped us to significantly improve our work. We are confident that the manuscript now shows significantly more clearly and convincingly the characteristics and capabilities of the proposed method. We gratefully thank the Reviewers for their feedback on our manuscript and for the stimuli provided to us.

Please, find below our detailed answers. Modifications to the paper with respect to the original submission have been highlighted in blue color.

Thank you for your time and your consideration.

Best regards,

Francesco Regazzoni, Stefano Pagani, Matteo Salvador, Luca Dede', Alfio Quarteroni

Reviewer # 1

The manuscript introduces a LD-net architecture, which is specifically designed to learn the spatial-temporal solutions of PDEs governed by first-order dynamics. This network employs two FCNs, a dynamic net and a reconstruction network, which function to map the input function to latent variables and subsequently map these latent variables to the comprehensive solution of the PDE. The proposed method's efficacy is then tested on the Advection-Diffusion-Reaction equation, Unsteady Navier-Stokes, and the Aliev-Panfilov electrophysiology model.

I found the paper to be a compelling read, well-articulated and furnished with clear and easy-to-follow mathematical problem/method definitions. However, my principal concern pertains to the novelty of the work. The LD-net method, proposed by the paper, is essentially the same to the Recurrent Neural Operator [1,2], that is developed for the study of homogenization problems. Although the contexts of application differ considerably, the underlying structure and key conclusions appear to be fundamentally similar.

This critique does not intend to undermine the present work, given that its application differs substantially from the preceding studies, and it's intriguing to observe a similar structure's effective application across diverse problems. Nevertheless, I believe it is essential for the authors to acknowledge the Recurrent Neural Operator, pointing out similarities and differences since this architecture is not novel to their work.

We thank the Reviewer for bringing the Recurrent Neural Operator (RNO) to our attention. RNO represents an approach of great interest, of which we were not aware. We see that LDNets and RNO share some similarities, which mainly lie in the presence of latent variables that evolve due to a neural network with a recurrent structure. However, the two methods have fundamental differences, which we illustrate in what follows.

In the RNO, the strain history at a given point is provided as input to the recurrent NN, and the evolution of the latent state is provided as input to a second NN, which provides the stress tensor at the point under consideration. The latent variables thus represent memory locally in space. In LDNets, instead, we have a single set of latent variables for the entire domain. The output of interest at a given point in space is obtained through a second neural network, in which the spatial variable is an input (the reconstruction neural network thus represents a field, and not, as in the RNO, point-wise evaluation). Therefore, while the latent variables of the RNO act locally in space, in LDNets they act globally. This is because the RNO represents a surrogate for dynamics at the microscale, whereas LDNets surrogate the system as a whole. This is reflected in important architectural differences: one of the key design choices of LDNets, which differentiates them from common approaches in the literature, lies in its meshless nature, which is not present in the RNO because of its local nature. In conclusion, LDNets and RNO are two very effective methods but with different purposes and scopes of application.

Therefore, we can state that RNO and LDNets are fundamentally two different methods. Still, because of the similarity in the way temporal dynamics is captured, we considered appropriate to add a reference to RNO in the manuscript (section Methods):

The latent variables $\mathbf{s}(t)$ allow to keep track of the state of the system. These, however, are not defined a priori (unlike methods based on dimensionality reduction techniques, see SI), but the latent space is discovered during the training process. This is similar to [9] for the case of time signals as outputs and to the Recurrent Neural Operator (RNO) [1, 6], used to learn microscopic internal variables capable of tracking the history dependence in multiscale materials. However, while in the RNO the latent variables correspond to a local material memory, LDNets have a single set of latent variables for the entire domain.

There are also some minor issues I'd like to raise:

Q1.1 It would be beneficial for the authors to consider moving portions of the LD section currently in the appendix to the main text, as it contains vital information on how the LD-net is constructed and trained.

We have moved to the central part of the manuscript some parts previously contained in the section Methods, reported in the appendix. Specifically, we have added details about the initial state, normalization layers, a-priori enforcement of physical knowledge, training algorithms, hyperparameter tuning. We believe that the current version of the manuscript is more complete and accessible to the reader, while still complying with the journal's guidelines, which require that the section Methods be reported in the appendix.

Q1.2 Test cases 1 and 2 indicate that the test error decreases as the number of latent variables increases. It would be interesting to know what happens when an excessive number of latent variables, say more than 5, are used. Will the error eventually converge to a constant, as is suggested by studies [1,2]?

As suggested by the Reviewer, we have investigated more deeply the impact of the number of latent states on the model generalization accuracy. We have considered two cases: finite solution manifold dimension and (virtually) infinite solution manifold dimension. More specifically, we have first considered Test Case 1b with a number of latent variables ranging from 1 to 5. The results are reported in the new Figure 2c and commented in the main text (section Results):

Finally, we train LDNets for increasing number of latent variables, ranging from 1 to 5 (Fig. 2c). As expected, the prediction accuracy significantly drops when going from 1 to 2 latent variables, that is when the intrinsic solution manifold dimension is reached. With more than 2 latent variables, it reaches a plateau, showing only a slight decrease due to increased model capacity, accompanied by a larger variance in the output.

Next, we considered Test Case 1c, wherein we extended the initial experiment from the original manuscript. In this case, we widened our investigation to include up to 7 latent variables and broadened the hyperparameter search space by increasing the range of neurons. The results are commented in the revised manuscript (section Results):

First, we set $f_{\max} = 0.5$ and we train LDNets for increasing number of latent states, from 2 to 7. Remarkably, as the number of latent states increases, LDNets discover more effective encodings, that reflect in an increasing prediction accuracy (Fig. 2f, blue line). Unlike Test Case 1b, where the intrinsic size of the solution manifold is 2 and this leads to a stagnation of the error, here the error decreases significantly even for higher numbers of latent variables. Still, for higher numbers of latent variables, we have a slowdown in the decreasing trend of the error, due to two factors: on the one hand, the finite size (100 samples) of the training set (see in this regard Fig. 2a), on the other hand, the optimizer that may not find the global minimum of the loss function. By increasing f_{\max} to 1 and 2, the FOM state gets less prone to be represented by a compact encoding, since the spectrum of the solution is wider. Prediction accuracy is indeed lower than in the case $f_{\max} = 0.5$, but it improves greatly by increasing the number of latent variables.

Q1.3 Studies [1,2] have also indicated that the latent dynamics that can be discovered from the data are constrained by the training dataset. Thus, critical latent dynamics can be missed if the training data do not encompass this information. Does the same hold true for the PDEs examined here?

The Reviewer is right: the effectiveness of the LDNet in capturing the underlying latent dynamics depends on the richness of the training dataset. The dependency

of the generalization on the dataset size is quantitatively analyzed in Test Case 2b (see Fig 2a), in which we observe that, as the number of samples increases, the error on the testing set gets lower.

We have stressed this aspect in the revised manuscript (section Results):

Indeed, because of the non-intrusive nature of LDNets, their ability to discover system dynamics is limited by the information contained in the training set, as it is common in data-driven model reduction/discovery methods [1, 6, 9]. Still, as the input space is covered more densely, LDNets are able to leverage that information as attested by the significantly decreasing test error.

In conclusion, I believe this paper has considerable potential. However, it requires a thorough review and subsequent refinement by the authors. Given my observations, I recommend a major revision before further consideration.

We thank again the Reviewer for the insightful suggestions and questions, which provided us with the opportunity to conduct a more profound investigation into our methodology, ultimately enhancing the quality of our manuscript.

Reviewer ¶ 2

The authors propose a network that utilizes 1) a neural ODE to propagate reduced order latent dynamics, and 2) a second network to map the low order dynamics to the high dimensional space. The approach is mesh-less, avoiding the use of autoencoders, as the grid-point to be evaluated is passed as an input to the network. As the proposed networks do not operate on the high dimensional space, they are faster compared to other alternatives in the literature.

The authors test their method on the advection diffusion reaction equation, where they demonstrate that LDNets can capture the dominant low-order Fourier modes, and improve its accuracy as the latent space is increased, in trivial test cases. LDNets are also benchmarked in the 2D lid-driven cavity flow, where the task is to predict the velocity field from the time-varying forcing. Last but not least, the LDNets are benchmarked to other standard approaches (autoencoders, LSTMs, etc.) in the Aliev-Panfilov (AP) model, which is a PDE describing the propagation of the electrical potential in an excitable tissue, demonstrating lower error.

The language of the paper is clear. The analysis is scientifically sound. In general the claims / conclusions of the paper are supported by the analysis. However, there are some methodological concerns.

We thank the Reviewer for carefully reading our manuscript and for the relevant comments and questions, which helped us to improve our work. Below are our detailed responses.

Q2.1 The novelty of the approach is the introduction of a reduced order latent space in the neural ODE, as all other main features (mesh-less, Neural ODEs, external forcing dependent network) of the method can be found in various other works (PINNs, etc.). How does the method

compare to just using a Neural ODE with inputs the external forcing and the grid point? Why does there have to be a latent space? The latent space is used by other methods for dimensionality reduction. It does not seem to be needed in this architecture, as the gridpoint is provided in the input.

The presence of a latent state is a crucial architectural choice for the proposed method. Indeed, the latent state allows the learned model to track the global state of the system and it allows nonlocal information to propagate across the computational domain. To demonstrate this, we have implemented the architecture suggested by the Reviewer, namely an ODE-Net fed by the input signal and the query point. We have tested this architecture in one of the test cases reported in the paper, and compared its results with those of LDNets (see section Results, Test Case 1c and the new Figure 3). The results revealed that, when increasing the number of latent states, LDNets are significantly more accurate in fitting the testing data and in generalizing to unseen inputs than the architecture without latent states here considered.

We have introduced these results and considerations in the revised manuscript (section Results):

To further assess the crucial role of including latent states in the model, we analyze the results obtained by removing the latent variables, namely by considering an ODE-Net fed by the input signal and the query point, and tracking the evolution of the output at the considered point (see SI for more details). We train this architecture by considering Test Case 1c, with $f_{\max} = 0.5$. To ensure a fair comparison, we employ the same dataset and the same hyperparameter tuning algorithm used for LDNets. The results (see Fig. 3) reveal that without latent states the prediction accuracy of the model is significantly reduced. Moreover, the greater the number of latent states, the greater the ability of the model to capture finer and finer features of the dynamics. We conclude that the presence of a latent state is a crucial architectural choice for LDNets. The latent variables allow nonlocal information to propagate across the computational domain Ω . With the architecture considered in this comparison, instead, the solution evolves in each point unaware of the state of surrounding points, despite the point coordinate is provided to the ODE-Net. Conversely, LDNets are able to learn systems whose dynamics is determined by spatial correlations. Notable examples are provided in the next sections.

We thank the Reviewer for suggesting this comparison.

Q2.2 In general the authors demonstrate that indeed LDNets can learn low order dynamics driven by an external forcing. However, the LDNets do not operate on the high dimensional space. While this characteristic is viewed as a strength of the method, it appears unlikely that the LDNets can adapt to scenarios where dynamics are driven by the initial condition or complex spatial correlations, rather than the forcing term, unless some modifications are incorporated. This is an important pitfall of the proposed approach.

We agree with the Reviewer that the proposed approach does not fit the case where the dynamics depends on a space-dependent initial condition and/or distributed forcings (i.e., fields). This is acknowledged in the manuscript (page 2) when we state that the input $u(t)$ needs to be “a set of time-dependent signals or constant parameters”. On the other hand, the proposed method is able to capture the evolution of systems whose dynamics is characterized by spatial correlations, thanks to the presence of the latent variables that provide a compact description of the system state, globally in space. This is demonstrated, e.g., by Test Case 2, where velocity fields are characterized by the presence of vortices, and by Test Cases 3 and 4, where, due to the presence of a traveling front, the value of the solution at a point is significantly determined by the value of the solution at surrounding points in previous times. Clearly, the number of required latent states will depend on how complex these spatial dynamics are, as demonstrated by Test Case 1c, where we vary the maximum frequency of the input precisely to make the spectrum of the solution manifold increasingly broad and thus more and more difficult to be captured with few latent variables.

In the new version of the manuscript, we have made it clearer that the type of inputs considered in this paper does not include initial conditions or input fields (section Discussion):

We have presented LDNNets, a novel class of NNs that learn in a data-driven manner the evolution of systems exhibiting spatio-temporal dynamics in response to external input signals.

[...]

A limitation of our work is that it does not consider the case of space-dependent inputs and of variable initial conditions, which will be the subject of future works; still, we remark that the class of problems that can be tackled with the proposed method encompass a broad range of real-life applications.

Indeed, we deem that this limitation does not undermine the potential impact of the proposed method, since the class of problems that can be tackled with the proposed method encompasses a broad range of real-life applications, i.e., all cases in which a space-time dynamic is driven by time-dependent control variables (which could be the power of a motor, the flow rate of a stream, the controlled release of a drug, the time-variation of the strength of a social measure, etc.).

Q2.3 The errors that the methods (LDNNets, AE+LSTMs, etc.) demonstrate in included benchmark (Aliev-Panfilov model) are very small, close to machine precision. From a machine learning standpoint, the task appears to be effectively solved by all methods. I am not sure about the relevance of demonstrating that the proposed approach reaches errors closer to machine precision compared to other works.

The test errors generated by the methods we are benchmarking against our proposed approach are on the order of 10^{-2} on average and 10^{-1} in the worst-case scenario, which is significantly far from machine precision. These error levels are indeed not negligible, as can be visually appreciated from Fig. 5d-e-f and the videos provided as Supplementary Materials, which show the presence of major

fluctuations in transmembrane potential, artifacts that significantly invalidate the quality of the results. In contrast, the solution produced by LDNets is free of such artifacts, and almost indistinguishable from the high-fidelity model solution. Similar comments apply to the newly introduced Test Case 4. Moreover, although these error levels may be considered very low in Data Science tasks, Scientific Computing problems typically require much higher levels of accuracy, making, in our opinion, the results of this work meaningful and of interest to the community.

Q2.4 In Figure 5 for example, all models fairly low errors. The differences most likely stem from architectural choices, and hyper-parameter optimization.

LDNets exhibit an NRMSE of $7.37 \cdot 10^{-3}$, whereas the other methods all register NRMSE values of $3.37 \cdot 10^{-2}$ or higher, which is nearly 4.5 times larger or more. Furthermore, the worst-case error over the testing set is even tenfold greater than that of LDNets. Hence, we believe that the differences are significant and indicative of the superior performance of LDNets. Moreover, we notice that, in all cases, the choice of hyperparameters comes from a thorough automatic selection procedure using state-of-the-art algorithms (see Section “Hyperparameter tuning algorithms”). During the hyperparameter tuning procedure, indeed, the hyperparameter space is explored by means of a Bayesian method, and a large number of architectural choices are considered. The reported results are obtained with the optimal hyperparameter configuration selected by the algorithm. In this way, each method is considered with an architecture that has been optimized using the same criteria and resources, ensuring the most fair comparison possible.

We have better stressed this important aspect in the revised version of the manuscript (section Results):

To ensure a fair comparison, we rely on an automatic tuning algorithm to select the optimal hyperparameter values for the different methods, setting an upper bound of $d_s \leq 12$ on the latent space dimension. The reported results are obtained with the optimal hyperparameter configuration selected by the tuning algorithm, independently for each method.

Q2.5 In Figure 4, it seems that the LDNets learn a latent space qualitatively very different compared to all other approaches. The latent space seems to be diverging exponentially. What is the intuition behind that? Is that a desirable characteristic?

We thank the Reviewer for their question, which stimulated a more in-depth analysis of the proposed method. We concluded that the behavior observed in Test Case 3 is not intrinsic to LDNets, but is due to the particular features of this test case. We have added a remark in this regard in the revised version of the SI (Section 3.3):

We observe that, in this test case, the latent variables of the LDNet diverge as time goes by, as the system under consideration is also characterized by divergent dynamics, associated with waves propagating

from their sources, with no return to the resting state. In contrast, in Test Case 2, wherein we consider time horizons long enough to contain the reversal in the flow directions, we observe latent variables that often return close to the origin (see Fig. 5).

We have thus introduced in the SI a new figure (Fig. 5) showing the time evolution of the latent variables in 6 different samples belonging to the test set of Test Case 2. As expected, in this case, when the input signal reverses its direction, we observe that many of the state variables also reverse the direction of their motion, possibly with some delay, thus accounting for inertial effects. Similarly, in Test Case 1 the dynamics of testing samples does not feature a divergent behavior, which is reflected by the latent states trajectories (see the new Fig. 3 of the main text).

Also, as noted by the Reviewer, LDNNs learn a latent space that qualitatively differs from that of the other methods. In particular, trajectories are more regular than for auto-encoder-based methods. We have introduced a paragraph commenting this aspect in the section Discussion:

We notice that the trajectories of the latent states $\mathbf{s}(t)$ obtained with LDNNs are smoother than those obtained with auto-encoder-based methods (see Fig. 5). This difference can be understood by considering how the latent state is constructed within auto-encoder-based methods. First, these methods learn a compact encoding of the high-dimensional output, thus defining a low-dimensional set of state variables, and then they attempt to find a law ruling their time evolution. However, while training the auto-encoder, the latent space is constructed with the sole purpose of allowing the output to be accurately reconstructed, without it necessarily being significant to the system dynamics. This issue is partially mitigated by a subsequent end-to-end training phase, which partially redefines the state variables in a way that is functional not only to reconstruct the solution, but also to capture the dynamics of the system. LDNNs, instead, thanks to the simultaneous training of the dynamics NN and the reconstruction NN, do not incur in this issue, as the training algorithm seeks the latent space that simultaneously pursue the twofold role of tracking the system dynamics and reconstructing the output at each time.

Q2.6 The latent dynamics are propagated through a neural ODE. These networks are notoriously hard to train, and many works attempt to alleviate the associated problems [1, 2, 3, 4]. Did the authors employ any mechanism to alleviate the training problems?

We have added a remark on this topic in the revised manuscript (section Methods):

Training ODE-Nets often presents challenges and typically involves an adaptive time integration to deal with stiff dynamics, which makes the computational graph potentially very deep and the computational cost often prohibitive [2, 3, 5, 7]. In this work, instead, we rely on a fixed time step size to integrate the latent variables. Thanks to the fact that the latent variables are not fixed a priori, but are defined at training

stage, the training algorithm tends to define latent variables with non-stiff dynamics, whose evolution is well captured through a fixed time step size, regardless of the stiffness of full-order model employed to generate the data. An evidence for this is provided by Test Case 3: the ground-truth model (Aliev-Panfilov) features, as it is well known, very stiff dynamics [4], thus imposing the use of a timestep of $5 \cdot 10^{-6}$ s, whereas the LDNet succeeds in fitting the results with great accuracy while using a much larger timestep (equal to $1 \cdot 10^{-3}$ s). This behavior is observed in our preliminary work [8] as well.

Q2.7 The authors do not provide results concerning the training time, and inference time (to represent the whole spatial field in testing) and comparisons with other methods.

We have reported in the revised manuscript the computational times associated with both the offline phase (namely, construction of the basis for both the solution manifold and DEIM for the POD/DEIM method, and parameters training for the other methods) and with the online phase (namely, the prediction of the solution for new inputs). The numerical values are reported in Table 1, and the associated comments are reported in the main text (section Results):

Furthermore, we observe that the POD-DEIM method results in a very limited speed-up with respect to the other methods considered. This limitation is intertwined with the necessity, due to numerical stability reasons, for the POD-DEIM model to be solved on the same temporal discretization as the high-fidelity model. This requirement represents a considerable constraint compared to the other methods outlined in this paper. As a matter of fact, as shown in Table 1, the computational cost for each sample with the FOM is about 37 s, the POD-DEIM method with 60 modes allows it to be reduced to about 8 s, when the other methods all lead to times less than 0.02 s. To this amount of time must be added the time required to evaluate the solution given the state variables, which, however, depends on the number of time steps and points at which this is required. For auto-encoder methods, the points at which this evaluation occurs are pre-established, taking $8.9 \cdot 10^{-7}$ s for each timestep. Conversely, LDNets, thanks to their mesh-less nature, offer the flexibility to evaluate at arbitrary locations, requiring $1.9 \cdot 10^{-7}$ s for each point in time and space. In this test case, should we want to evaluate the solution at all time steps and training points, this would correspond to about $4.5 \cdot 10^{-4}$ s for auto-encoder-based methods and $9.5 \cdot 10^{-3}$ s for LDNets. That said, we observe that, with the exception of POD/DEIM, the inference times associated with the other methods are virtually negligible compared to the time required to evaluate the Full Order Model (FOM).

Concerning the offline time, associated with model construction, the training cost of LDNets (22 887 s) is lower than that of auto-encoder-based methods, except for AE/LSTM (11 009 s), which, however, yields a poor accuracy in the predictions. In fact, the accuracy achieved by AE/LSTM is matched by LDNets after just 1 354 s of training. On the other hand, the accuracy levels of AE/ODE and AE/ODE-e2e are

attained by LDNet after 6510 and 9090 s, respectively. The POD-DEIM method, as expected, is characterized by a less heavy offline phase, which, however, does not lead to a speed-up comparable to the other methods in evaluation.

Moreover, we have introduced an additional test case (Test Case 4), to further compare the proposed method with auto-encoder-based methods. In this test case as well, we have reported training and inference times (see Table 2), and we have commented them in the text:

The trained model inference time (online time) is comparable among the three methods considered. The time required to reconstruct the solution from the latent states for auto-encoder-based methods is $1.1 \cdot 10^{-4}$ s per time instant, while for LDNet it is $3.9 \cdot 10^{-6}$ s per time and space point. In all cases, the methods considered lead to a remarkable speedup with respect to the time required by the FOM (807 s per simulation). As for the offline stage, the time required to complete training with the three models is similar (nearly between 75 000 and 95 000 s). The LDNets, however, achieve higher levels of accuracy in less time. In fact, the accuracy achieved with AE/ODE is reached by LDNet after only 3 569 s, while that of AE/ODE-e2e after 4 530 s.

We thank the Reviewer for this suggestion.

Q2.8 Publishing the code and data would assist reproducibility and strengthen the claims of the paper.

After publication, the codes will be made publicly available. For the moment, the Reviewers can access the code at the URL:

https://polimi365-my.sharepoint.com/:f:/g/personal/10377072_polimi_it/E18g8HrJYahFgdY0FDzUkgsBnAzpjAH-I6rSJao-cxMQ0g?e=f9fcLx

To download the files, enter the password:

D4#Bzt63*2uq

References (by Reviewer # 1)

- [1] Bhattacharya, K., Liu, B., Stuart, A., & Trautner, M. (2023). Learning Markovian homogenized models in viscoelasticity. *Multiscale Modeling & Simulation*, 21(2), 641-679.
- [2] Liu, B., Ocegueda, E., Trautner, M., Stuart, A. M., & Bhattacharya, K. (2023). Learning macroscopic internal variables and history dependence from microscopic models. *Journal of the Mechanics and Physics of Solids*, 105329.

References (by Reviewer # 2)

- [1] Xuanqing Liu, Tesi Xiao, Si Si, Qin Cao, Sanjiv Kumar, and Cho-Jui Hsieh. Neural sde: Stabilizing neural ode networks with stochastic noise. *arXiv preprint arXiv:1906.02355*, 2019.

- [2] Chris Finlay, Jörn-Henrik Jacobsen, Levon Nurbekyan, and Adam Oberman. How to train your neural ode: the world of jacobian and kinetic regularization. In *International conference on machine learning*, pages 3154–3164. PMLR, 2020.
- [3] Marin Biloš, Johanna Sommer, Syama Sundar Rangapuram, Tim Januschowski, and Stephan Günnemann. Neural flows: Efficient alternative to neural odes. *Advances in neural information processing systems*, 34:21325–21337, 2021.
- [4] Arnab Ghosh, Harkirat Behl, Emilien Dupont, Philip Torr, and Vinay Namboodiri. Steer: Simple temporal regularization for neural ode. *Advances in Neural Information Processing Systems*, 33:14831–14843, 2020.

References

- [1] K. Bhattacharya, B. Liu, A. Stuart, and M. Trautner. “Learning Markovian homogenized models in viscoelasticity”. In: *Multiscale Modeling & Simulation* 21.2 (2023), pp. 641–679.
- [2] E. Dupont, A. Doucet, and Y. W. Teh. “Augmented Neural ODEs”. In: *Advances in neural information processing systems* 32 (2019).
- [3] C. Finlay, J.-H. Jacobsen, L. Nurbekyan, and A. Oberman. “How to train your neural ODE: the world of Jacobian and kinetic regularization”. In: *International conference on machine learning*. PMLR. 2020, pp. 3154–3164.
- [4] P. C. Franzone, L. F. Pavarino, and S. Scacchi. *Mathematical cardiac electrophysiology*. Vol. 13. Springer, 2014.
- [5] A. Ghosh, H. Behl, E. Dupont, P. Torr, and V. Namboodiri. “Steer: Simple temporal regularization for Neural ODE”. In: *Advances in Neural Information Processing Systems* 33 (2020), pp. 14831–14843.
- [6] B. Liu, E. Ocegueda, M. Trautner, A. M. Stuart, and K. Bhattacharya. “Learning macroscopic internal variables and history dependence from microscopic models”. In: *Journal of the Mechanics and Physics of Solids* (2023), p. 105329.
- [7] X. Liu, T. Xiao, S. Si, Q. Cao, S. Kumar, and C.-J. Hsieh. “Neural SDE: Stabilizing Neural ODE networks with stochastic noise”. In: *arXiv preprint arXiv:1906.02355* (2019).
- [8] F. Regazzoni, L. Dedè, and A. Quarteroni. “Active contraction of cardiac cells: a reduced model for sarcomere dynamics with cooperative interactions”. In: *Biomechanics and Modeling in Mechanobiology* (2018), pp. 1–24.
- [9] F. Regazzoni, L. Dedè, and A. Quarteroni. “Machine learning for fast and reliable solution of time-dependent differential equations”. In: *Journal of Computational Physics* 397 (2019), p. 108852.

REVIEWERS' COMMENTS

Reviewer #1 (Remarks to the Author):

The authors have addressed all my concerns and there is no reason to further delay the acceptance of this paper.

Reviewer #2 (Remarks to the Author):

The authors have addressed all comments.

Congratulations on the meticulous work.